

# Enhancing Long-Term Trend Simulation of Global Tropospheric OH and Its Drivers from 2005-2019: A Synergistic Integration of Model Simulations and Satellite Observations

**Amir H. Souri[1,2]\*, Bryan N. Duncan[1], Sarah A. Strode[1,2], Daniel C. Anderson[1,3], Michael E. Manyin[1,4], Junhua Liu[1,2], Luke D. Oman[1], Zhen Zhang[5,6], and Brad Weir[2,7]**

[1]Atmospheric Chemistry and Dynamics Laboratory, NASA Goddard Space Flight Center (GSFC), Greenbelt, MD, USA
[2]GESTAR II, Morgan State University, Baltimore, MD, USA
[3]GESTAR II, University of Maryland Baltimore County, Baltimore, MD, USA
[4]Science Systems and Applications, Inc., Lanham, MD, USA
[5]National Tibetan Plateau Data Center (TPDC), State Key Laboratory of Tibetan Plateau Earth System, Environment and Resource (TPESER), Institute of Tibetan Plateau Research, Chinese Academy of Sciences, Beijing, China
[6]Earth System Science Interdisciplinary Center, University of Maryland, College Park, MD, USA
[7]NASA Global Modeling and Assimilation Office (GMAO), Goddard Space Flight Center, Greenbelt, MD, USA

\* Corresponding author: a.souri@nasa.gov



## Abstract

The tropospheric hydroxyl radical (TOH) is a key player in regulating oxidation of various compounds in Earth's atmosphere. Despite its pivotal role, the spatiotemporal distributions of OH are poorly constrained. Past modeling studies suggest that the main drivers of OH, including $NO_2$, tropospheric ozone ($TO_3$), and $H_2O(v)$, have increased TOH globally. However, these findings often offer a global average and may not include more recent changes in diverse compounds emitted on various spatiotemporal scales. Here, we aim to deepen our understanding of global TOH trends for more recent years (2005-2019) at 1×1 degrees. To achieve this, we use satellite observations of HCHO and $NO_2$ to constrain simulated TOH using a technique based on a Bayesian data fusion method, alongside an interpretable machine learning module named ECCOH, which is integrated into NASA's GEOS global model. This innovative module helps efficiently predict the convoluted response of TOH to its drivers/proxies. Aura Ozone Monitoring Instrument (OMI) $NO_2$ observations suggest that the simulation has high biases over biomass burning activities in Africa and Eastern Europe, resulting in overestimation of up to 20% in TOH, regionally. OMI HCHO primarily impacts oceans where TOH linearly correlates with this proxy. Five key parameters including $TO_3$, $H_2O(v)$, $NO_2$, HCHO, and stratospheric ozone can collectively explain 65% of variance in TOH trends. The overall trend of TOH influenced by $NO_2$ remains positive, but it varies greatly because of the differences in the signs of anthropogenic emissions. Over oceans, TOH trends are primarily positive in the northern hemisphere, resulting from the upward trends in HCHO, $TO_3$, and $H_2O(v)$. Using the present framework, we can tap the power of satellites to quickly gain a deeper understanding of simulated TOH trends and biases.

## 1. Introduction

The hydroxyl radical (OH) regulates the lifetimes of a vast number of key atmospheric compounds, such as sulfur dioxide ($SO_2$), nitrogen dioxide ($NO_2$), volatile organic compounds (VOCs), carbon monoxide (CO), and methane ($CH_4$). Despite its outsized importance for atmospheric chemistry and climate, our knowledge on both the abundance and long-term trends of OH is limited due to its sparse observations, manifesting in large discrepancies between simulated OH among global models (e.g., Naik et al., 2013; Zhao et al., 2019; Murray et al., 2021; Fiore et al., 2024). Particularly, these discrepancies can introduce large uncertainties when it comes to precisely representing methane (Holmes et al., 2013; Nguyen et al., 2020), a potent greenhouse gas. Consequently, to understand the potential impact of this warming agent on climate shifts and extreme weather events, it is essential to accurately simulate methane concentration within a coupled climate model, such as the NASA's Goddard Earth Observing System (GEOS) model (Molod et al., 2015; Nielsen et al., 2017), which requires reasonable representation of its major sink – reaction with OH.

Despite the challenges posed by OH's short lifespan of less than two seconds, low-pressure laser-induced fluorescence spectroscopy has proven invaluable in measuring OH for over twenty airborne field campaigns (Miller and Brune, 2020). These datasets have been instrumental in verifying the efficacy of chemical mechanisms involving varying reaction rate coefficients and aerosol heterogeneous chemistry (Brune et al., 2019; Miller and Brune, 2020; Brune et al., 2022), understanding urban air quality (Brune et al., 2022; Souri et al., 2023), as well as identifying potential sources of $HO_x$ ($OH+HO_2$) that may have been hampered due to instrument detection limits and/or unmeasured compounds (e.g., Ren et al., 2008). However, while these observations offer valuable insights, they are limited in time and space and cannot provide a full picture of tropospheric OH abundance.

There are several approaches that have been employed to constrain OH needed for replicating observed values of a tracer whose primary sink is OH and its sources are relatively well known. One notable method is methyl chloroform (MCF) inversion (Patra et al., 2014; Turner et al., 2017; Rigby et al., 2017;





Naus et al., 2019). However, this method only provides hemispheric-average OH and is thus insufficient to
resolve the spatial distribution of OH.
A more sophisticated approach to constraining OH is to incorporate well-characterized satellite
observations of factors known to influence OH, such as $NO_2$, CO, ozone, and formaldehyde (HCHO), into
a chemical transport model using inverse modeling and/or chemical data assimilation methods (Sandu and
Chai, 2011; Bocquet et al., 2015). This method offers a crucial advantage in that it accounts for the
interconnectedness of various chemical and physical processes within model increments. For example,
adjustments to $NO_x$ levels will impact nitrate and ozone concentrations, which in turn affect the $HO_2$ uptake
through aerosols, OH, and radiation, reciprocally leading to a more accurate representation of $NO_x$. Several
studies have used subsets of satellite observations to improve $HO_x$ and ozone chemistry, with Miyazaki et
al. (2020) using a diverse range of observations, including CO, $NO_2$, $O_3$, and nitric acid ($HNO_3$), to improve
model predictions using a local ensemble Kalman filter. The incorporation of these observations led to a
reduction in the asymmetric OH ratio between the northern and southern hemispheres, aligning better with
MCF results (Patra et al., 2014). Similarly, Souri et al. (2020a) leveraged well-characterized observations
of HCHO and $NO_2$ to improve ozone chemistry over East Asia using non-linear analytical Bayesian
inversion, observing significant changes in OH levels after adjusting biogenic VOC in southeast Asia.
While incorporating these observations into atmospheric models offers a comprehensive way to gain
insights into spatiotemporal OH variability, it is complicated by several layers of complexity, such as
unidentified satellite biases, unresolved scales in satellite observations, and errors in models including
transport, chemical mechanisms, vertical diffusion, and depositions rates. Understanding how these errors
could cloud the realistic determination of OH requires running constrained models under various
realizations, which is computationally prohibitive.
Researchers have developed OH predictors based on a set of key parameters, offering reasonable
spatial and temporal coverage without compromising computational efficiency (Spivakovsky et al., 2000;
Duncan et al., 2000; Elshorbany et al., 2016; Nicely et al., 2018; Wolfe et al., 2019; Nicely et al., 2020;
Anderson et al., 2022, Zhu et al., 2022; Anderson et al., 2023; Baublitz et al., 2023). These studies fall into
four categories, the first of which uses box model photochemical simulations to predict OH levels under a
steady-state assumption, using a blend of pre-modeled fields and various observations influencing OH
(Spivakovsky et al., 2000; Nicely et al., 2018). The second group uses proxy observations (e.g., HCHO or
water, $H_2O$) of OH in remote areas (Wolfe et al., 2019; Baublitz et al., 2023). The third group employs
high-order polynomials to establish an empirical relationship between OH and different parameters,
avoiding the need to solve numerous differential equations in chemical mechanisms (Duncan et al., 2000;
Elshorbany et al., 2016). Finally, the fourth group leverages powerful machine learning algorithms to
encapsulate the complexities between OH and its key influencers to efficiently predict OH using a
comprehensive dataset which is easily exchangeable between models (Nicely et al., 2020; Anderson et al.,
2022; Zhu et al., 2022; Anderson et al., 2023).
In this work, we demonstrate the potential of a new approach to constrain simulated OH that uses
satellite observations to adjust the input parameters to an improved parameterization of OH (Anderson et
al., 2022), within the Efficient $CH_4$-CO-OH (ECCOH) (pronounced "echo") configuration (Elshorbany et
al., 2016) of NASA's GEOS model. We use the Modern-Era Retrospective analysis for Research and
Applications, Version 2 (MERRA2) reanalysis data (Molod et al., 2015) to constrain meteorology and
adjust two critical OH inputs using the latest Aura Ozone Monitoring Instrument (OMI) $NO_2$ and HCHO
retrievals (Lamsal et al., 2021; Nowlan et al., 2023) from 2005-2019 worldwide. Through conducting a
range of experiments, we determine the extent to which leveraging OMI $NO_2$ and HCHO observations can
enhance current representations of these two species derived from a global model simulation, MERRA2-
GMI (hereafter M2GMI) (Strode et al., 2019), so that we can achieve more accurate portrayals of OH
abundance and its long-term trends. Ultimately, we deconvolve the intricate OH trend maps into five critical
parameters using various modeling experiments, including tropospheric ozone, stratospheric ozone, $NO_2$,
HCHO, and $H_2O$.
Our paper is structured into several sections. In sections 2.1 to 2.3, we discuss the model
configurations, Bayesian data fusion algorithm, and satellite observations used. In section 2.4, we outline





our modeling experiments, which aim to uncover the impact of various key OH inputs on its trends and assess the effect of OMI adjustments. In section 3.1, we examine the discrepancies between our prior knowledge from M2GMI and OMI observations and demonstrate how the data fusion can mitigate these differences. In section 3.2, we delve into the effect of OMI adjustments to $NO_2$ and HCHO on tropospheric OH (TOH) magnitudes across the globe. In section 3.3, we focus on understanding the long-term effect of a set of key inputs on OH and how well they can replicate our most dynamic representation of TOH. In Section 4, we summarize the potential of using satellite observations in conjunction with well-characterized models to identify biases and long-term trends in TOH and discuss the limitations of our current analysis and potential paths forward.

## 2. Models, Methods, and Measurements

### 2.1. Models

#### 2.1.1. GEOS

The GEOS model (Molod et al., 2015; Nielsen et al., 2017) simulates global weather with 1° longitude × 1° latitude spatial resolution. The model follows 72 hybrid sigma values ranging from the surface to 0.01 hPa. We employ a cumulus parameterization to consider deep convection (Moorthi and Suarez, 1992). Cloud microphysics is determined by a single-moment parameterization based on Bacmeister et al. (2006). We activate the "replay" option (Orbe et al., 2017) to constrain several meteorological variables using the MERRA-2. Sea surface temperatures and ice content are pre-described from various observations (Nielsen et al., 2017; Reynolds et al., 2007). Speciated aerosol concentrations and their optical properties are simulated by the GOCART model (Chin et al., 2002) within GEOS. The rapid radiative transfer model for GCMs (RRTMG) resolves the long- and short-wave radiation imposed by GOCART-simulated aerosols, allowing for the direct impact of aerosol on meteorology to be taken into consideration (Nielsen et al., 2017). The period of simulation starts in 2005 and ends in 2020. Ten years before 2005 are considered for the spin-up of meteorological, CO, and $CH_4$ fields.

#### 2.1.2. ECCOH

A computationally-efficient module, named ECCOH was developed to simulate the chemistry of the $CH_4$-CO-OH cycle in the GEOS-5 model framework (Elshorbany et al., 2016). CO and $CH_4$ tracers are explicitly simulated and their emissions are discussed in Sections 2.1.2.1 and 2.1.2.2. A key component of ECCOH is the parameterization of tropospheric OH, which was developed using a gradient boosted regression tree machine learning algorithm (Anderson et al., 2022) and is a function of chemical, solar irradiance, and meteorological variables. The training dataset of chemical and meteorological variables was a 40-year daily M2GMI model simulation (Strode et al. 2019), which includes tropospheric chemistry involving 120 species and 400 reactions with the GMI mechanism (Duncan et al., 2007a and the references therein) and uses MERRA-2 reanalysis to constrain transport and meteorology at 0.625×0.5 degrees.

We present the variables used as inputs to the parameterization of OH for this study in Table 1. The daily archived chemical inputs are from the M2GMI simulation with several variables being constrained with observations. For instance, both $NO_2$ and HCHO fields are corrected whenever satellite observations are available as described in Section 2.2.1. We chose $NO_2$, an observable compound from satellites and a reasonable proxy for $NO_x$ that has been shown to affect OH (e.g., Zhao et al., 2020; Anderson et al., 2022). HCHO is used as a proxy for VOC oxidation via OH in remote oceanic regions (Wolfe et al., 2019).



There are also long-term satellite data records of other OH drivers, including water vapor
(e.g., Aqua AIRS) and total ozone column (e.g., Aura OMI), that we could also consider. However,
the GEOS MERRA-2 system already assimilates satellite datasets of water vapor and the M2GMI
simulation simulates well (i.e., <4%) the total ozone column as compared to observations (Figure
S1). The integrated water vapor columns from MERRA2 and microwave-based satellite
observations over-ocean also agree well (<5%), especially after 2000 when many satellite
observations have been used in the reanalysis data (Figure 3 in Bosilovich et al., 2017). Therefore,
the application of the "replay" mode constrains various meteorological fields, providing a more
realistic reconstruction of OH studied here.
Tropospheric ozone is another critical input to the parameterization of OH. Although we
will compare M2GMI tropospheric ozone with satellite observations to locate any differences,
reliable measurements of tropospheric ozone from satellites are lacking due to the limited
sensitivity of the retrievals to ozone in low altitudes. Therefore, our study refrains from imposing
any observational constraint on tropospheric ozone.

### 2.1.2.1. Monthly CO emissions

We use a modified version of EDGAR (Emissions Database for Global
Atmospheric Research) v5.0 (Crippa et al., 2019), which is a comprehensive database that
provides estimates of sector-based CO emissions from human activities (i.e.,
anthropogenic) on a global scale. Previous studies (e.g., Zheng et al., 2019) suggested a
large underestimation of EDGAR CO emissions for India and China. Accordingly, we
scale up the residential and transportation emissions from China by a factor of 1.6, and the
residential emissions from India by a factor of 1.2 based on Zheng et al. (2019). The
emissions spanned the entirety of the study period, from 2005 until 2020, and were
prepared monthly at a spatial resolution of $0.1^\circ \times 0.1^\circ$. The daily biomass burning emissions
are CMIP6 emissions, which derived from on the Global Fire Emissions Database version
4 with small fires (GFED4s) (van Marle et al., 2017). To account for the chemical
production of CO from the oxidation of non-methane VOCs, we adopt the CO yield
estimates from Duncan et al. (2007b) (i.e., a molar yield of 20% from isoprene, 20% from
monoterpenes, 100% from methanol, 67% from acetone, 19% from anthropogenic VOC
emissions, and 11% from biomass burning VOC sources) and released these CO emissions
in the first vertical level of the model. With regards to the biogenic VOC emissions used
for the above CO production estimates, we use offline MEGAN calculations using a
GEOS-Chem (v13.2.0) run. CO production from $CH_4$ oxidation is calculated online for
each model box.

### 2.1.2.2. Monthly $CH_4$ emissions

In this study, several bottom-up $CH_4$ emissions related to anthropogenic, wetland,
natural, and biomass burning sources are used to simulate $CH_4$. The monthly-basis
anthropogenic sources are derived from EDGARv6 (Ferrario et al., 2021). The biomass
burning emissions come from the GFED4s. Because EDGARv6 accounts for agricultural
waste burning, we exclude this specific source from the GFED4. Following Strode et al.
(2020), we use modified monthly-basis natural emissions from ocean, termite, and mud
volcano emissions. Wetland emissions are derived from an improved dynamic wetland
emission framework at $0.5^\circ \times 0.5^\circ$ based on the TOPography-based hydrological model
(TOPMODEL) (Zhang et al., 2016; Zhang et al., 2023). A climatological sink of $CH_4$ from
soil uptake is subtracted from the total $CH_4$ emissions.



**Table 1**. The list of inputs used for the parametrization of OH.

| Input Group | Variables (**Directly Constrained**) | Source | Temporal Resolution |
|---|---|---|---|
| Offline Chemical Species | **NO₂**, **HCHO**, O₃, isoprene, acetone, H₂O₂, propene, propane, methyl hydroperoxide, ethane, C4 and C5 alkanes, and stratospheric O₃ columns | M2GMI (offline) (Strode et al. 2019) | Daily-averaged |
| Online Chemical Species | CO and CH₄ | GEOS (online) | Daily-averaged |
| Meteorological Fields | **T**, **P**, **Qv**, and cloud fraction | GEOS (online) | Daily-averaged |
| Optical Properties | Aerosol optical depth; ice crystal cloud optical depth; and water droplet cloud optical depth at above and below of a given model vertical layer. | GEOS (online) | Daily-averaged |
| Geographic Information | Latitude and solar zenith angle (SZA) | Calculated | Fixed for latitude, but daily for SZA based on local noontime |
| Surface Properties | Surface UV albedo | OMI LER climatology (Qin et al., 2019; Fasnacht et al., 2019) | Monthly (climatology) |


***2.2. Methods***
*2.2.1. Bayesian data fusion for NO₂ and HCHO fields using OMI retrievals*

To improve the representation of M2GMI NO₂ and HCHO concentrations and their long-
term trends, which are used as input to the parameterization of OH in ECCOH, we scale their
columnar mass using Aura OMI observations of NO₂ and HCHO columns (described in Sections
2.3.1 and 2.3.2) using an offline version of the optimal interpolation (OI) method (Parish and
Derber, 1992; Jung et al., 2019) with an appropriate regularization. If we assume that the error
covariances of M2GMI columns and OMI ones follow a Gaussian distribution with zero means and
their relationships are linear, we can estimate new columns using Bayes' theorem (Rodgers, 2000):

$$\mathbf{X}_a = \mathbf{X}_b + \gamma\mathbf{BH}^{\mathrm{T}}(\gamma\mathbf{HBH}^{\mathrm{T}} + \mathbf{E})^{-1}(\mathbf{Y} - \mathbf{HX}_b) \tag{1}$$

where $\mathbf{X_b}$ is the prior M2GMI columns (i.e., background), $\mathbf{X}_a$ is the posterior M2GMI columns
(i.e., analysis), $\mathbf{B}$ is the error covariance matrix of the a priori, $\mathbf{E}$ is the error covariance matrix of
the observations, $\mathbf{Y}$ is the observations, and $\mathbf{H}$ is the observational operator which is equivalent to
the identity matrix in our case. $\mathbf{E}$ is populated by the average sum of precision error squares the
satellite product provides. We interpolate both $\mathbf{E}$ and $\mathbf{Y}$ into the M2GMI grid box using a mass-
conserved linear barycentric interpolation method. The National Meteorological Center's (NMC)
approach is a common technique for calculating $\mathbf{B}$ in atmospheric models (Parish and Derber 1992;



Souri et al., 2020b); however, due to computing constraints, rerunning the M2GMI model to create
the 24-hour prediction segments needed in the NMC method was not possible. Instead, we initialize
**B** by setting it to 50% errors for $NO_2$ and HCHO, both of which are subject to regularization. $\gamma$ is
the regularization factor designed for achieving the best fit (minimum residuals between **Y** and
**HX$_b$**) while minimizing the effect of the noise in the observations (minimum variance in **X$_a$**). To
this end, we seek an optimal regularization factor based on finding the "knee point" in the curve of
the incremental regularization factors (ranging from 0.1 to 10) and the degrees of freedom obtained
from the optimization. The $\gamma$ value is determined based on the average of all data points in a month
and does not vary from pixel to pixel to ease the interpretation of the result. We did not account for
the non-diagonal spatial correlations of **B**, as it requires us to carry out the NMC method. We use
the ratio of **X$_b$**/**X$_a$** to uniformly scale the three-dimensional concentrations of the target gas (i.e.,
$NO_2$ or HCHO). The error associated with the constrained M2GMI columns can be obtained via

$$\mathbf{S_a} = (\mathbf{I} - \gamma \mathbf{B}\mathbf{H^T}(\gamma \mathbf{H}\mathbf{B}\mathbf{H^T} + \mathbf{E})^{-1}\mathbf{H}) \times \gamma \mathbf{B} \tag{2}$$

The averaging kernels (AK) describe the amount of information gained from the observations are
represented by

$$\mathbf{AK} = \mathbf{I} - \frac{\mathbf{S_a}}{\mathbf{B}} \tag{3}$$

where **I** is the identity matrix.
In our research, we have created an open-source Python package called OI-SAT-GMI
(Souri, 2024), which possesses the ability to download and process OMI level 2 products, perform
air mass factor (AMF) recalculation, and conduct mass-conserved interpolation, while also
executing the OI algorithm.
In our approach, the adjustments are implemented to the M2GMI output (i.e., a data fusion
approach instead of data assimilation one), thereby restricting the full use of improved $NO_2$ and
HCHO representation for more accurate simulation of other chemical compounds impacted by $NO_2$
and HCHO, including ozone (e.g., Souri et al., 2020a, 2021). Nevertheless, as the accuracy of $NO_2$
concentrations can significantly impact OH and HCHO is strongly tied to VOC oxidation through
OH in remote ocean areas (Wolfe et al., 2019), the adjustments are expected to be beneficial in
achieving a more robust representation of OH.
*2.2.2. Trend analysis*
We determine a linear trend in a time series based on fitting the following equation
accounting for a seasonal cycle and shorter frequencies in the observations:

$$\boldsymbol{y} = a_0 + a_1\boldsymbol{t} + \sum_{i=1}^{3} a_{i+1} cos2\pi\omega_i(\boldsymbol{t} - \varphi_i) \tag{4}$$

The equation comprises several variables, including $a_0$ as the mean, $a_1$ as the linear trend, $\boldsymbol{t}$ as time,
$a_{i+1}$, $\omega_i$, and $\varphi_i$ are the amplitude, frequency, and phase, respectively. We consider three
harmonics ($\omega_i = 1,2,3$) to account for seasonal cycle ($\omega=1$) and higher frequencies. To assess the
statistical significance of a trend, we employ the Mann-Kendall test and consider a trend to be
significant if the linear trend passes the test at a 95% confidence level.
In the context of trend analysis, the careful examination of errors in observations ($\boldsymbol{y}$) is a
critical aspect often overlooked. However, when the errors of observations are obtainable, such as
those obtained from satellites or constrained M2GMI fields, we determine the parameters by
applying a weighted estimation. This estimation is optimized using a Levenberg–Marquardt
algorithm. Considering the errors in the observational data deemphasizes more uncertain data,
resulting in a more realistic determination of the linear trend.



### 2.2.3. OH response calculations

To elucidate the response of OH to different input parametrizations, such as $NO_2$, HCHO, and $O_3$, we determine the semi-normalized sensitivities through a traditional finite difference method:

$$SOH_i = \frac{[OH]_i^{110\%} - [OH]_i^{90\%}}{0.2} \tag{5}$$

where $[OH]_i^{110\%}$ and $[OH]_i^{90\%}$ are OH concentrations from perturbing input parameters ($i$) by 1.1 and 0.9 scaling factors in the ECCOH offline framework (Anderson et al., 2022).

### 2.3. Measurements

### 2.3.1. OMI MINDS tropospheric $NO_2$ columns

To improve the representation of $NO_2$ fields used as input to the parameterization of OH, we constrain the archived monthly fields with the most updated NASA standard tropospheric $NO_2$ product (v4.0; Lamsal et al., 2021) from Aura OMI. Aura has a local equatorial overpass time of 13:45 and nearly daily global coverage. This new OMI product version is improved in multiple aspects as compared to the former products, including surface reflectance and cloud retrieval (Lamsal et al., 2021).

The validation of OMI tropospheric $NO_2$ columns from the comparison to integrated aircraft spirals obtained from diverse air quality campaigns revealed a good level of correlation ($r>0.7$) (Choi et al., 2020). However, large mean biases, approximately 40%, were observed. These biases come from various sources, including systematic biases in prognostic data utilized in the retrieval, biases inherent in the aircraft data, spatial representation errors (Judd et al., 2020; Souri et al., 2022), and temporal representation errors. The spatial representation errors have been recognized to notoriously drift the slopes from the unity line in validation studies (Souri et al., 2022). Notably, Choi et al. (2020) achieved a substantial reduction in mean biases, decreasing from 40% to 16%, through the downscaling of OMI data into a finer resolution domain using a regional chemical transport model. Likewise, Pinardi et al. (2020) reduced the biases between MAX-DOAS and OMI $NO_2$ observations by considering a radial dilution factor to account for the mismatch scales between the satellite footprint and the pointwise observations. These studies showed that the true statistics describing OMI biases are unknown, but they tended to be milder than those derived from directly comparing large pixels with pointwise measurements. It is important to highlight that discrepancies between M2GMI and OMI $NO_2$ will surpass the reported biases, thereby underscoring the product's reliability over diverse geographical regions.

The long-term trends of tropospheric $NO_2$ columns have undergone extensive comparative analyses with in-situ observations (Lamsal et al., 2015; Pinardi et al., 2020), regulatory inputs, and assessments of human and biomass burning activities (Duncan et al., 2016; Choi and Souri, 2015a,b; Krotkov et al., 2016; Jin and Holloway, 2015; Souri et al., 2017; Rueter et al., 2014; de Foy et al., 2016; Hickman et al., 2021).

We prefer level 2 over level 3 products to enable the recalculation of AMFs with time-varying shape factors derived from the M2GMI simulation. We removed low-quality pixels using the main quality flag, cloud fraction >30%, terrain reflectivity >20%, and those pixels affected by the "row anomaly" complication. The data product, which has a spatial resolution ranging from ~13 km × 24 km (at nadir) to ~24 km × 160 km (at extremities of the scanline), were then regridded to the M2GMI grid (0.625°×0.5° degrees) using a mass-conserved linear barycentric interpolation method. The AMF recalculation was performed via:

$$VCD_{new} = \frac{VCD_{old} AMF_{old}}{AMF_{new}} \tag{6}$$



where $VCD_{old}$ and $AMF_{old}$ are the default states of tropospheric vertical columns and air mass
factors. $AMF_{new}$ is determined by summing the product of scattering weights and the M2GMI partial
columns from the surface to the tropopause level prescribed in the OMI level 2 data.
*2.3.2. OMI SAO total HCHO columns*

For the same reason as OMI $NO_2$, we use OMI SAO total columns based on a newly-
developed algorithm framework by Nowlan et al. (2023). The new retrieval represents a major step
forward in the surface albedo treatment including the bidirectional reflectance distribution function
for land (BRDF) from the MODIS product (MCD43C1 Version 6.1) extended to the UV
wavelengths using a principal component algorithm. Since there are no MODIS BRDF data
available over water, the algorithm uses the Cox-Munk slope distribution to estimate the surface
reflectance of water bodies (Cox and Munk, 1954). An important issue with the long-term record
of OMI HCHO measurements is the artificial increasing trend brought on by sensor degradation
(Choi and Souri, 2015a,b, Gonzalez Abad et al., 2015). The algorithm uses an earthshine spectrum
over the Pacific Ocean with a latitudinal and solar zenith-dependent correction factor described in
Nowlan et al. (2023) to mitigate this artifact.
The new SAO algorithm has been validated with Ozone Mapping and Profiler Suite
(OMPS) data radiance with respect to Fourier-transform Infrared Spectroscopy (FTIR) in-situ
measurements in 2012-2020, showing a relative bias of 30% based on monthly-averaged data
(Kwon et al., 2023). While the validation results based on the OMI radiance have not been released
yet, it is likely for the biases to stay at roughly the same range of errors at monthly-gridded OMI
data onto the M2GMI grid which is comparable to the OMPS footprint (50 km).
Once again, we used Eq.6 to recalculate OMI HCHO total columns with dynamical shape
profiles produced during the M2GMI simulation. We remove unwanted pixels using the following
criteria: the main quality flag, cloud fraction >40%, and flag for pixels affected by the row anomaly.
We then regridded the data to the MERRA-2 GMI grid using the same approach used for OMI
$NO_2$.
## 2.4. Experiments

We perform a series of experiments to investigate the sensitivity of OH to geophysical
variables known to influence or to be tied with OH. Table 2 lists all sensitivity experiments along
with their purposes and differences from an analysis (i.e., constrained) experiment. The pillar of all
experiments is the analysis experiment (*Sanalysis*) which uses i) chemical variables from a full-
chemistry simulation as input to the parameterization of OH in ECCOH (Section 2.1.2; Table 1);
ii) transport and metrological fields constrained by MERRA2 reanalysis data (Section 2.1.1); iii)
long-term estimates of monthly CO and $CH_4$ emissions (Section 2.1.2.1 and 2.1.2.2); iv) optical
depths of clouds and aerosols along with observed climatology of OMI UV surface albedo; and v)
the $NO_2$ and HCHO fields constrained by the Bayesian data fusion method (Section 2.2.1).
To examine the importance of having $NO_2$ and HCHO fields constrained with OMI data,
we design three experiments imitating *Sanalysis,* but withholding the OI scaling factors one at a
time. We then subtract these model outputs from those of *Sanalysis* and name them as *SOMInitro*,
*SOMIform*, and *SOMInitroform*.
The other experiments are intended to systematically isolate the chemical effect of a
specific driver/proxy of OH trends. Due to the significant impact of $NO_2$, tropospheric ozone,
stratospheric ozone column, and water vapor on the primary or secondary pathways of OH
loss/production (Naik et al., 2013; Murray et al., 2013; Strode et al., 2015; Nicely et al., 2018; Zhao
et al., 2020; Anderson et al., 2021), we include four experiments (*SOHwv*, *SOHnitro*,
*SOHtropozone*, and *SOHstratozone*) to single out each effect on OH trends. Additionally, we
include HCHO (*SOHform*), a robust proxy for VOC oxidation via OH in remote ocean regions



(Wolfe et al., 2019) to understand how those chemical pathways have changed over time. In these
experiments, we set the target driver constant to the monthly values in the first year of simulation,
and subsequently subtract these model outputs from *Sanalysis*. Amongst various OH
drivers/proxies considered, water vapor exclusively comes from the GEOS online simulation; to
isolate the water vapor effect on OH only, we provide fixed water vapor fields from MERRA2
based on the monthly-varying 2005 simulations. Simultaneously, GEOS is allowed to simulate
water vapor online to address meteorology.
Using ambient gas concentrations in the ECCOH model poses a challenge in distinguishing
the respective factors contributing to their variations. For instance, it is difficult to discern the
distinct influences of lightning-produced $NO_2$ versus anthropogenic $NO_2$ on the abundance of OH.
However, an advantageous feature of our approach is that various observational sources constrain
the data fields used via the Bayesian data fusion method or MERRA2 reanalysis data.

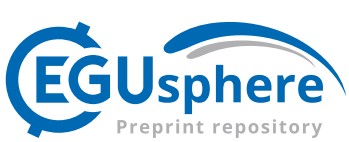

**Table 2.** The experiments designed to assess the effect of various OH drivers/proxies and OMI constraints on TOH trends and magnitudes.

| Model Scenario | Term | Difference from the analysis run | Purpose |
|---|---|---|---|
| Analysis (*constrained*) | *Sanalysis* | -- | The "best effort" to simulate the evolution of the CH$_4$-CO-OH cycle from 2005-2019. |
| Analysis –[a] defaulting to NO$_2$ M2GMI | *SOMInitro* | Uses archived M2GMI monthly-averaged NO$_2$ concentration fields. | Isolate the importance of constraining M2GMI NO$_2$ concentration fields with OMI observations. |
| Analysis – defaulting to HCHO M2GMI | *SOMIform* | Uses archived M2GMI monthly-averaged HCHO concentration fields. | Isolate the importance of constraining M2GMI HCHO concentration fields with OMI observations. |
| Analysis – defaulting to NO$_2$ and HCHO M2GMI | *SOMInitroform* | Uses archived M2GMI monthly-averaged NO$_2$ and HCHO concentration fields. | Isolate the importance of constraining M2GMI NO$_2$ and HCHO concentration fields with OMI observations. |
| Analysis – fixed H$_2$O vapor | *SOHwv* | The dynamical water vapor fields fed to the parameterization of OH are fixed to the monthly-varying 2005. | Isolate the impact of the long-term trend of water vapor on OH. |
| Analysis – fixed tropospheric ozone | *SOHtropozone* | M2GMI ozone fields are set to the monthly-varying 2005. | Isolate the impact of the long-term trend of tropospheric ozone burden on OH. |
| Analysis – fixed NO$_2$ | *SOHnitro* | M2GMI NO$_2$ fields are set to the monthly-varying 2005. | Isolate the impact of the long-term trend of NO$_2$ on OH. |
| Analysis – fixed HCHO | *SOHform* | M2GMI HCHO fields are set to the monthly-varying 2005. | Understand the long-term trend of HCHO strongly tied with VOC oxidation via OH in remote regions. |
| Analysis – fixed stratospheric ozone column | *SOHstratozone* | M2GMI stratospheric ozone field fed to the parameterization of OH is set to the monthly-varying 2005. | Isolate the impact of the long-term trend of stratospheric ozone columns on OH. |

[a] "–" denotes the subtraction operator.



## 3. Results and Discussion

### 3.1. Spatial distributions and trends analysis of several inputs to the parameterization of OH

We begin our analysis with an examination of the long-term trends and magnitudes of two key inputs (HCHO and $NO_2$) to the parameterization of OH. Some other key parameters, such as total ozone columns, tropospheric ozone columns, and water vapor are also shown in Figure S1-3, Figure S7-8, and Text S1.

### 3.1.1. Tropospheric $NO_2$ columns

We performed two sets of comparisons; the first comparison involves examining the differences in the tropospheric $NO_2$ columns in the M2GMI relative to those of OMI before and after applying the OI correction. The second comparison focuses on the global 2-D maps of long-term linear trends of OMI, M2GMI prior to and after the Bayesian data fusion correction synched at the satellite viewing condition.

Figure 1 demonstrates the absolute difference in M2GMI tropospheric $NO_2$ columns with respect to those of OMI before (the a priori) and after (the a posteriori) the data fusion application along with AK in 2005-2019. In-land regions show positive biases over several regions, including central Africa (box A), the Midwest U.S. (box B), and Europe (box C). The same tendency was observed in Anderson et al. (2021). The largest contributor to $NO_2$ in box A and box C is biomass burning activities (Jaeglé et al., 2005; Giglio et al., 2012), suggesting that either the emission factors and/or the total dry mass burnt were possibly too high in these regions.

M2GMI overestimates $NO_2$ concentrations in non-urban areas in box B which tend to be more severe during summertime. Although soil $NO_x$ emissions could be the first explanation for this phenomenon, accounting for about 30% of tropospheric $NO_2$ columns in the region according to Vinken et al. (2014), the soil $NO_x$ parameterization used in M2GMI relies on Yienger and Levy (1995), which is known to have a low bias (Jaeglé et al., 2005; Hudman et al., 2012; Vinken et al., 2014; Souri et al., 2016). Therefore, there may be other uncertainties in the model concerning chemistry (e.g., Canty et al., 2015) or area anthropogenic $NO_x$ emissions (Hassler et al., 2016) causing the bias.

A large portion of metropolitan areas in the Middle East, Europe, and the U.S. shows an underestimation of $NO_2$ in M2GMI. Moreover, OMI observations reveal large positive biases over the North China Plain (NCP), a region exhibiting exceptionally high $NO_2$ levels (e.g., Duncan et al., 2016; Krotkov et al., 2016; Souri et al., 2017). This is primarily because of not accounting for the recent aggressive emissions mitigation in China in the bottom-up emission inventory used in the model. We observe several regions over China and Yellow Sea underestimating $NO_2$ with respect to OMI observations that do not improve considerably after the adjustments. This tendency is a result of the use of a fractional error for populating the error covariance matrix of the a priori, rendering the prior error too low. Although we used a regularization factor to battle this problem, it did not vary from region to region. A regionally-adaptive regularization factor could be a possible remedy for this problem but at a cost of overcomplicating the interpretation of the results.

Expectedly, the Bayesian fusion greatly mitigates the regional biases, with notable reductions observed over central Africa, China, the U.S., Amazon, and Europe. The regional biases (>80%) well exceed the reported biases associated with OMI tropospheric $NO_2$ product (<40%), suggesting that the adjustments should be considered as improvement. Nonetheless. it is important to acquire an abundance of long-term records from surface spectrometers such as MAX-DOAS and Pandora to comprehensively evaluate the degree of enhancement of M2GMI constrained by OMI within the troposphere, which is currently unavailable for the period of 2005-2019 to our best knowledge. The reduction in the biases over remote areas in the tropics is less noteworthy due to large errors in the observations. In other words, it is difficult to have high confidence in the degree of deficiency the model can have in simulating $NO_2$ over pristine areas by comparing it to OMI. This notion mathematically manifests in low AK in remote areas showing that rich information from OMI tropospheric $NO_2$ gravitates more towards polluted regions.



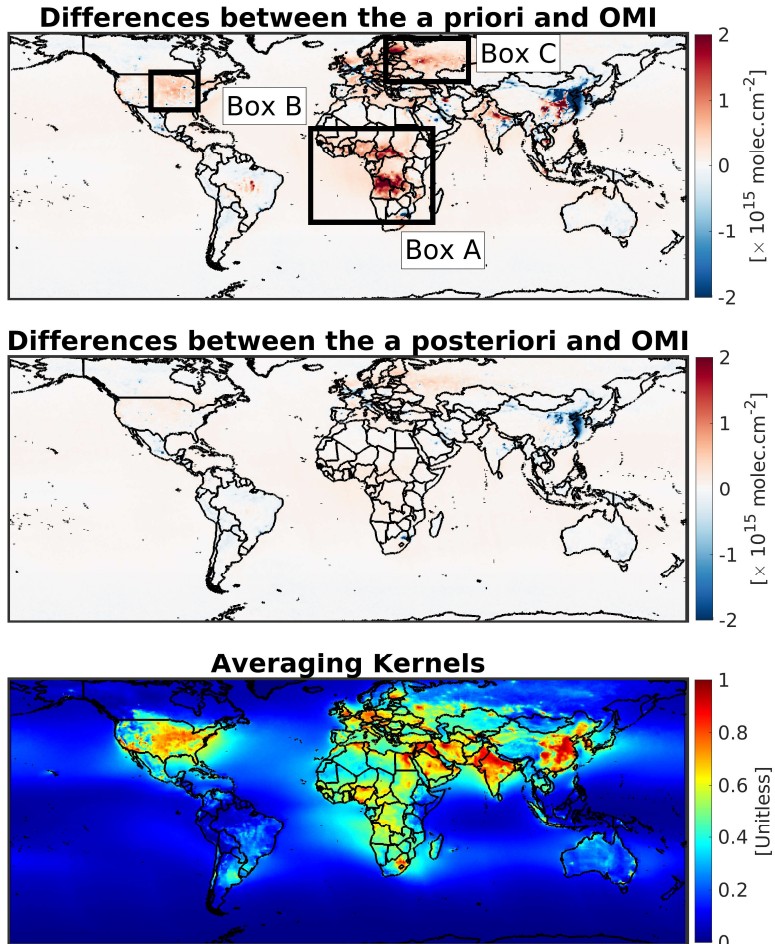

**Figure 1**. The global maps of M2GMI tropospheric $NO_2$ annual difference with respect to those of OMI before applying the Bayesian data fusion correction factors (top) and after (middle) in 2005-2019; the mean of averaging kernels describing the gained information from OMI (bottom). Grids in high latitudes are removed from the figure due to too few numbers of samples OMI provided.

Figure 2 illustrates the linear trends of tropospheric $NO_2$ between 2005 and 2019 observed by OMI and simulated by the M2GMI before and after using the OI algorithm. The errors in OMI observations and the constrained M2GMI are considered while calculating the trends. Focusing on the trends by OMI, we observe a consistent picture compared to former studies (Duncan et al., 2016; Choi and Souri, 2015a,b; Krotkov et al., 2016; Jin and Holloway, 2015; Souri et al., 2017). High income countries, such as the U.S., those located in the western Europe, and major cities in Russia, undergo a significant reduction of $NO_2$ concentrations due to the implementation of emission mitigation regulations. Additionally, low and moderate income countries, such as those in the Middle East, northern Africa, and India, have seen upward trends in $NO_2$. Various signs of trends are observed in East Asia. Due to recent effective regulations in China (Zhang et al., 2012), we observe downward trends in the NCP region (Rueter et al., 2014; de Foy et al., 2016; Souri et al., 2017). The downward trend predominantly starting from 2011-2012 counteracts the upward trend in prior years resulting in



statistically insignificant linear trends. Both Japan and South Korea show downward trends during the period
of 2005-2019 (Duncan et al., 2016; Souri et al., 2017).

Encouragingly, the model prior simulation of the tropospheric $NO_2$ trend is consistent with OMI over
most of the polluted regions except for China, where the bottom-emission inventories used in the M2GMI fail
to reflect recent mitigation efforts occurring in NCP region. The posterior estimation is in a higher degree of
agreement compared to OMI (Text S2). An encouraging observation arising from the comparison of the
M2GMI prior with the posterior $NO_2$ trends is the achievement of a higher spatial variance (information) in
low and medium income countries (e.g., India and Iran). This finding suggests that the emission inventories
used in the M2GMI lacked adequate spatial information even at the model spatial resolution.

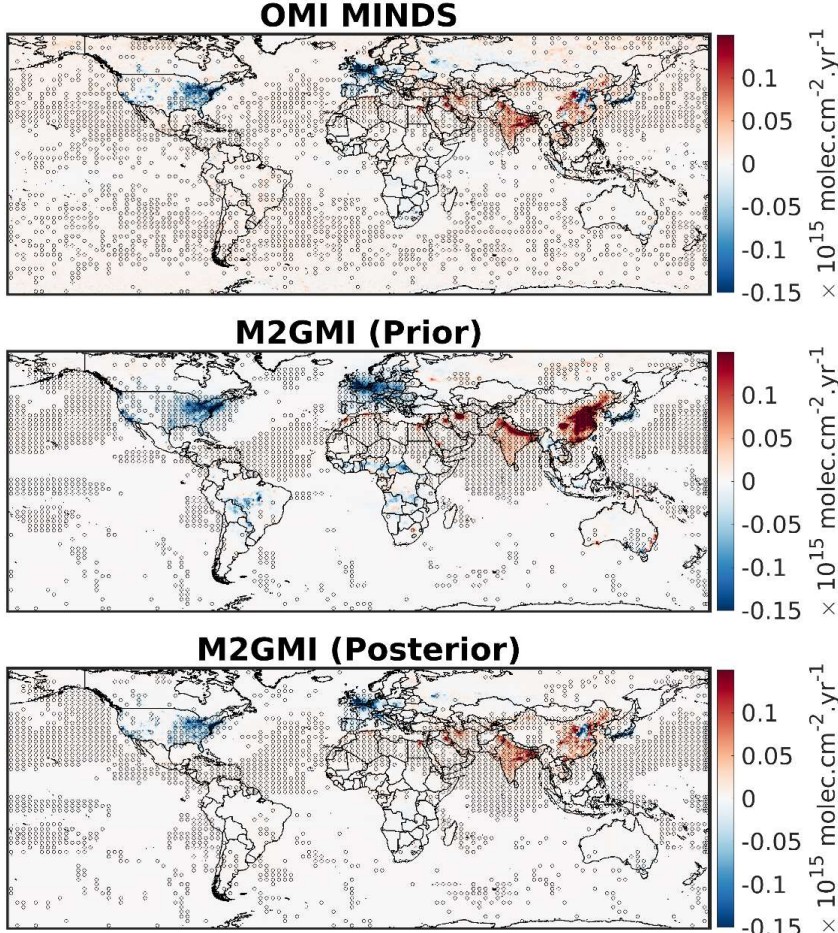

**Figure 2.** The global maps of linear trends of annual tropospheric $NO_2$ columns observed by OMI and
simulated by M2GMI before and after using the Bayesian fusion. The model simulations are sampled at the
exact time and location of OMI, and masked if OMI observations were unavailable due to data quality criteria
used. The dots indicate statistically significant trends at 95% confidence interval.



### 3.1.2. Total HCHO columns

We validate the simulated HCHO concentrations, drawing inspiration from the $NO_2$ comparison framework. Figure 3 illustrates the absolute differences in simulated HCHO total columns with respect to OMI before and after the Bayesian data fusion application, in addition to AK. The prior model simulation has considerable skill in capturing the HCHO total columns over several areas, such as the Middle East, Europe, India, and East Asia. However, marked positive biases are discernible in regions with abundant isoprene emissions, such as the Amazon, southeast Asia, southeast U.S., and central Africa. This outcome is most likely due to an overestimation of biogenic emissions; various investigations have reported a predominantly positive bias (between a factor of 2 to 3) linked to isoprene emissions estimated by the Model of Emissions of Gases and Aerosols from Nature (MEGAN) using satellite measurements in isoprene-rich regions (e.g., Millet et al., 2008; Stavrakou et al., 2009; Marais et al., 2012; Bauwens et al., 2016; Souri et al., 2020a).

The simulated HCHO concentrations are relatively too low over pristine areas, such as high latitudes and over mountains. This may be attributed to an underestimation of $CH_4$ in M2GMI because of assigning its values as background conditions (Strode et al. 2019). The integration of OMI satellite data has proven effective at reducing the biases in areas where HCHO concentrations are large because the signal-to-noise ratio tends to be large resulting in high AKs. Nonetheless, there are some adjustments over remote areas. In fact, OMI HCHO columns provide more information than OMI $NO_2$ in remote areas because background HCHO concentrations are not extremely low due to evenly distributed methane and methanol concentrations. It is worth noting that the biases in M2GMI well exceed the expected OMI HCHO column biases, suggesting that the adjustments to HCHO improve the model.

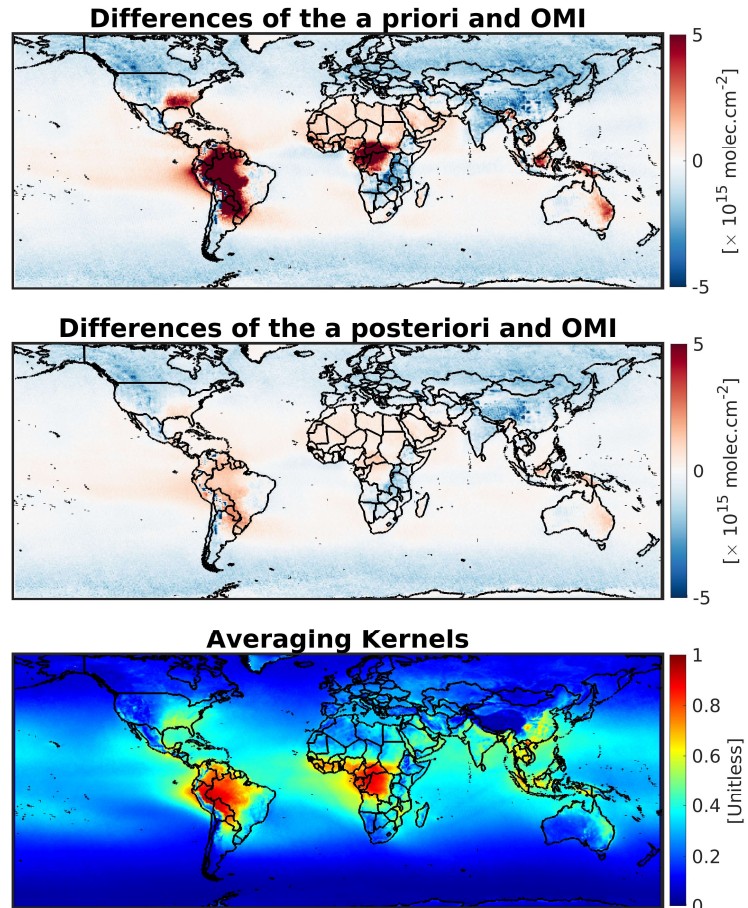

**Figure 3.** Same as Figure 1 but for HCHO total columns.

Figure 4 shows the global maps of HCHO total column trends derived from OMI, the prior M2GMI, and the posterior M2GMI. The widespread upward trends in HCHO over India are evident due to lack of effective efforts on cutting emissions related to volatile organic compounds (e.g., De Smedt et al., 2015; Kuttippurath et al., 2022; Bauwens et al., 2022). We observe HCHO columns going up in the northwestern US and over oil sands in Canada, possibly due to increased evergreen needleleaf forest and an increase in crude oil production (Zhu et al., 2017), respectively. The downward trends over the southeast US could be due to a decrease in drought events (Figure S5), which significantly affect isoprene emissions and the oxidation of VOCs (Duncan et al. 2009; Naimark et al., 2021; Wang et al., 2022). Alternatively, this downward trend could be partially due to the dampened HCHO production from VOC oxidation due to reduced $NO_x$ emissions (Marais et al., 2014; Wolfe et al., 2016; Souri et al., 2020c). In agreement with previous studies (Stavrakou et al. 2017, Souri et al., 2017, Shen et al., 2019, Souri et al., 2020a), HCHO columns increase over the NCP. HCHO columns tend to decrease over parts of central Africa (e.g., Democratic Republic of the Congo) and the Amazon basin potentially due to reduced deforestation rates (De Smedt et al., 2015; Jones et al., 2022). However, a large variability in the sign of HCHO trends over these regions is seen; Congo shows an opposite trend in comparison to that of Democratic Republic of the Congo; the northern portion of the Amazon basin increased. Encouragingly, the prior knowledge captures the upward trends over India and China along with





downward trends over central Africa. However, the magnitudes and spatial features of these trends are not
entirely in line with respect to OMI.
We do not fully understand HCHO trends over oceans. Part of these patterns might be caused by
transport from nearby sources. For instance, areas around south Asia, South America, and Gulf of Mexico are
affected by the trends over the land in their proximity. However, trends over several areas, such as the southern
part of the Indian Ocean, Australia, and Sahara, are not fully explainable by nearby sources. It is possible that
certain patterns can be linked to climate variability; for instance, there is growing evidence of more cyclonic
circulation intensifying westerly trade winds from central Africa due to warming Indian Ocean (Dhame et al.,
2020) that may contribute to rising HCHO. An in-depth understanding of HCHO trends over oceans certainly
deserves a separate follow-up study.
The posterior estimates better line up with the OMI trends, especially over the Amazon, India, and
Central Africa (Text S3). The correction factors, however, worsen the trends over the southeast US and
Canada. This is essentially due to the use of the fractional errors in the a priori making the OMI corrections
more impactful (i.e., higher Kalman gain) in summertime than in wintertime.

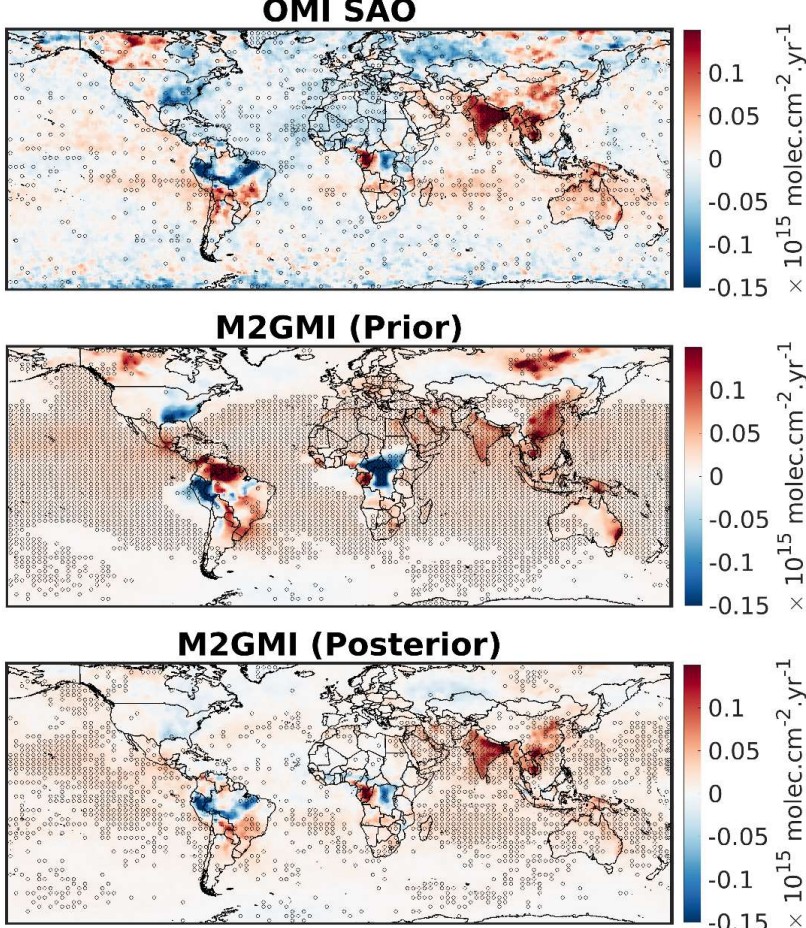

**Figure 4.** Same as Figure 2 but for total HCHO columns. The linear trends in OMI SAO are smoothed by a
median filter for better visualization.




In sum, we saw that M2GMI $NO_2$ and HCHO, both inputs to the parameterization of OH, were broadly
better presented through the integration of OMI observations. Consequently, the improvement is expected to
elevate the level of reliability in the experimental outcomes, particular in the context of *SOHnitro* and
*SOHform* simulations. As for other important compounds, such as stratospheric columns, tropospheric $O_3$, and
water vapor, the comparison of the model with OMI total ozone columns shows a strong degree of agreement
(<4% biases) with no significant trend in low-mid latitudes (Figure S1 and S2). The well-documented upward
trend in tropospheric ozone in the northern hemisphere is well reproduced by M2GMI (Figure S3). We did not
validate GEOS water vapor simulations, because of the use of MERRA2 reanalysis, which is thoroughly
validated in Bosilovich et al. (2017). Furthermore, the comparison of integrated water vapor linear trends from
our GEOS-5 run (2005-2019) with satellite data presented in Borger et al. (2022) shows a remarkable
agreement (Figure S7-8).
### *3.2. Added value of OMI on simulated tropospheric OH*
Here, we present the results from three OMI-related experiments (*SOMInitro*, *SOMIform*,
*SOMInitroform*) to understand the effect of OMI adjustments made to M2GMI on TOH. Throughout the paper,
TOH is determined based on the methane-reaction-weighted OH suggested by Lawrence et al. (2001).
Moreover, we calculate the response of TOH to $NO_2$ and HCHO using Eq.5.
Figure 5 consists of three columns, illustrating the percentage adjustments made by OMI $NO_2$ using
OI, the response of TOH to $NO_2$ concentrations, and the simulated TOH derived from the *SOMInitro*
experiment. The observed pattern of increments aligns with the improvements seen in Figure 1, with positive
(negative) values indicating underestimation (overestimation) of M2GMI. Broadly, the overestimates
dominate over underestimates resulting in the global tropospheric $NO_2$ reduction by ~4%. Upon segregating
the increments into four distinct seasons, it becomes evident that the adjustments do not uniformly apply to
every season. This non-uniformity is primarily attributed to biases in M2GMI, influenced by biomass burning
(box A, C) (Section 3.1.1), both of which exhibit strong seasonality.
Deciphering the precise chemical processes influencing the response of OH to $NO_2$ using a machine-
learning approach is challenging. However, it is widely recognized that $NO_x$ has positive feedback on OH
through increased $NO+HO_2$ and ozone (Murray et al., 2021; Zhao et al., 2020; He et al., 2021). Considering
$NO_2$ as a surrogate for $NO_x$, similar tendencies are expected, as evident from the positive numbers from the
sensitivity results obtained from offline calculations. The response of TOH to $NO_2$ displays a pronounced
seasonal cycle stemming mainly from photochemistry.
The impact of adjustments made by OMI $NO_2$ on TOH is most substantial over regions where both
the adjustments and TOH responses to $NO_2$ are significant. For instance, the large adjustments made over
Europe in DJF do not substantially affect TOH because the response value is low due to reduced
photochemistry.
On a global scale, changes to TOH are much milder (1% reduction) than those occurring regionally.
For instance, we see substantial regional impacts (up to 20%) over many areas such as Central Africa, the
Midwest US, the Middle East, and Eastern Europe. In light of the global reduction in OH, we observe global
column average methane mixing ratios ($XCH_4$) to increase by 10 ppbv on average (Text S4). This
augmentation happens monotonically with an increase of 0.9 ppbv per year, ultimately resulting in ~15 ppbv
difference at the end of the simulation (Figure S13). This is essentially due to the long lifetime of $CH_4$.
Likewise, the TOH reduction results in column average CO mixing ratio (XCO) enhancements which transpire
more locally than $XCH_4$ does due to the shorter XCO lifetime. The XCO enhancements reach above 10 ppbv
in Africa (Text S5).




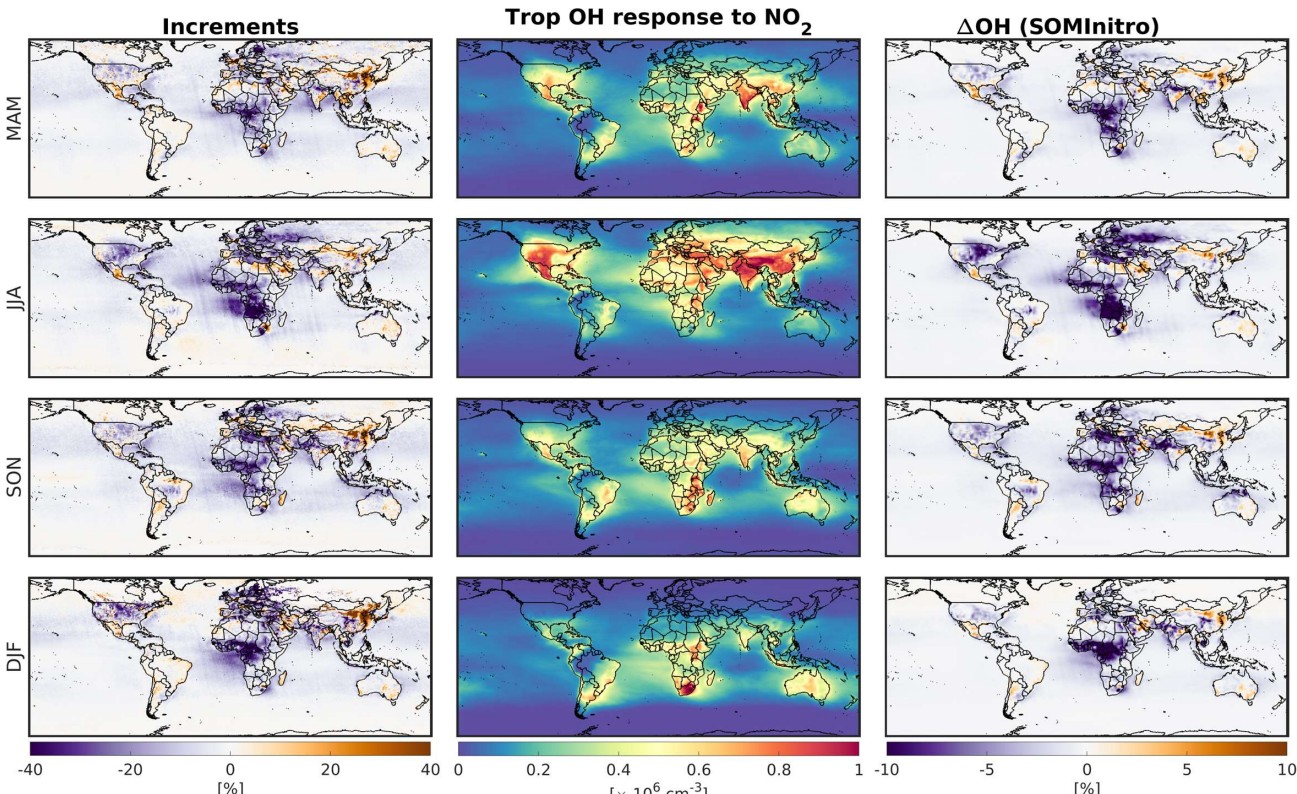

**Figure 5.** (first column) the percentage of adjustments applied to M2GMI $NO_2$ fields within the troposphere suggested by OMI tropospheric $NO_2$ columns for four different seasons, (second column) the semi-normalized response of tropospheric OH to tropospheric $NO_2$ changes based on ECCOH offline calculations, and (third column) the resulting effect of the adjustments on tropospheric OH derived from the online simulation (*SOMInitro*). MAM, JJA, SON, and DJF are acronyms for March-April-May, June-July-August, September-October-November, and December-January-February.

Figure 6 demonstrates the same scheme as Figure 5 but with a focus on the *SOMIform*. Marked negative increments are found in regions characterized by elevated isoprene concentrations because of the overestimations of M2GMI biogenic isoprene emissions. Positive increments are mostly confined to high latitudes and certain areas of East Asia (Section 3.1.2).

The interplay between HCHO and OH is contingent on the intricate dynamics governing HCHO production from the oxidation of VOCs and methane and HCHO loss from various chemical pathways (Valin et al., 2016; Wolfe et al., 2019). In remote areas where $HO_x$ is low, the prevailing sink of HCHO is through photolysis. Conversely, in more polluted areas, the reaction of HCHO+OH emerges as a competing loss pathway. Assuming a steady-state approximation, which is a reasonable assumption for pristine areas, the photolysis loss of HCHO dominates over the reaction with OH, resulting in a linear relationship between HCHO and OH. In other words, high (low) HCHO concentrations are indicative of high (low) TOH. It is because of this that we use HCHO as a proxy of TOH in remote oceans regions. In regions characterized by heightened $HO_x$ levels, OH and HCHO become decoupled. Encouragingly, our implicit parametrization of OH has considerable skill at elucidating these intricate chemical tendencies; specifically, it reveals muted



responses in regions with relatively tangible pollution levels, whereas positive responses are evident in oceanic
regions. Like results obtained for NO$_2$, the response map has a seasonal cycle due to photochemistry.
Because of the muted response of TOH to HCHO over land, a substantial portion of geographical
regions undergoing significant adjustments made by OMI becomes less important. TOH primarily changes
over oceanic areas in a way that it decreases in low latitudes but increases in high latitudes. The largest
reduction occurs in Amazon downwind where both increments and responses display large magnitudes. As a
result of these changes, we see a marginal increase in XCH$_4$ over tropics where OMI increments reduced TOH.
The HCHO adjustment did not noticeably affect XCO either (Text S5).
Modifications on HCHO by OMI do not signal substantial changes in background VOC oxidation
through OH. In fact, TOH changes by this proxy are of an order of magnitude less than those by OMI NO$_2$.
This tendency is a result of two key factors: i) the adjustments wield their major influence over oceans where
M2GMI has a fair performance, and ii) the amount of information obtained from OMI HCHO (i.e., AK)
remains somewhat limited in remote areas due to low signal-to-noise ratios.
Due to the rather independent nature of the TOH responses to NO$_2$ and HCHO, where the former
prevails over land and the latter over ocean, the concurrent adjustments of HCHO and NO$_2$ using OMI (i.e.,
*SOMInitroform*) results in a rather linear combination of outcomes derived from *SOMIform* and *SOMInitro*
(Figure S21). This linear outcome is characterized by a large decrease in TOH in low latitudes and a moderate
increase in high latitudes resulting in a decrease of global TOH by ~1%.

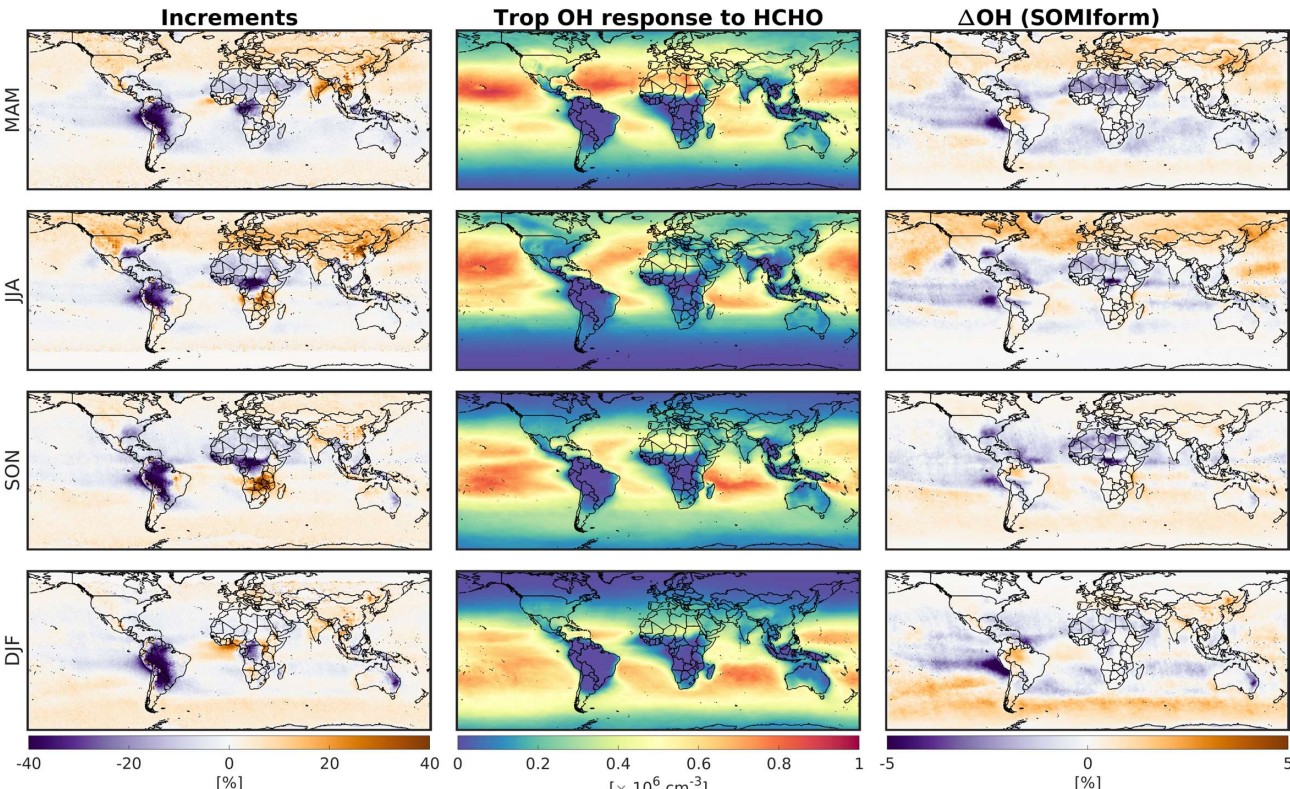

**Figure 6.** Same as Figure 5 but for HCHO.



### 3.3. Synergy of the model and satellite observations to explain TOH long-term trends

*3.3.1. The dominant contributor to TOH trends*

Here, we take advantage of the wealth of information from satellites and our well-characterized model used for the inputs to the parameterization of OH to rank the dominant contributor to TOH linear trends. By assuming that TOH follows a linear combination of each individual experiment designed to isolate OH driver/proxy (i.e., *SOHnitro*, *SOHform*, *SOHtropozone*, *SOHstratozone*, and *SOHwv*), wherein second (or higher) chemical feedback is disregarded, we can determine the biggest contributor to the TOH trend for each model grid box by finding which driver/proxy holds the largest absolute amount. We only label a grid if the absolute linear trend of the dominant driver/proxy surpasses the second most dominant one by 30%.

Figure 7 illustrates the dominant factor explaining TOH trends. Several patterns can be found from this result: i) $NO_2$ plays a significant role in TOH trends in various polluted areas, such as Asia and the Middle East; ii) the upward trend of TOH over the western Pacific Ocean is primarily attributed to increased tropospheric ozone from Asia (e.g., Lin et al., 2017); also, we observe a significant fraction of TOH over the tropical Atlantic Ocean increasing because of rising tropospheric ozone from Africa and Central/South America (Edwards et al., 2003); iii) HCHO is convolved with TOH trends over tropical oceans); iv) water vapor plays a pivotal role in shaping TOH trends over oceans across the globe; iv) stratospheric ozone columns are mostly significant over the South Pole due to the ozone healing process (Figure S2). The next sections will focus on the magnitude of these trends and the degree to which they can collectively explain the variance in TOH trends compared to *Sanalysis*.

It is important to recognize that the analysis presented here should be interpreted as a relative assessment of a limited number of TOH drivers/proxies, rather than an exhaustive evaluation of all the physical and chemical processes that are tied to TOH. Nonetheless, the data presented offers valuable insights into the TOH trends and can be used as a basis for further research.

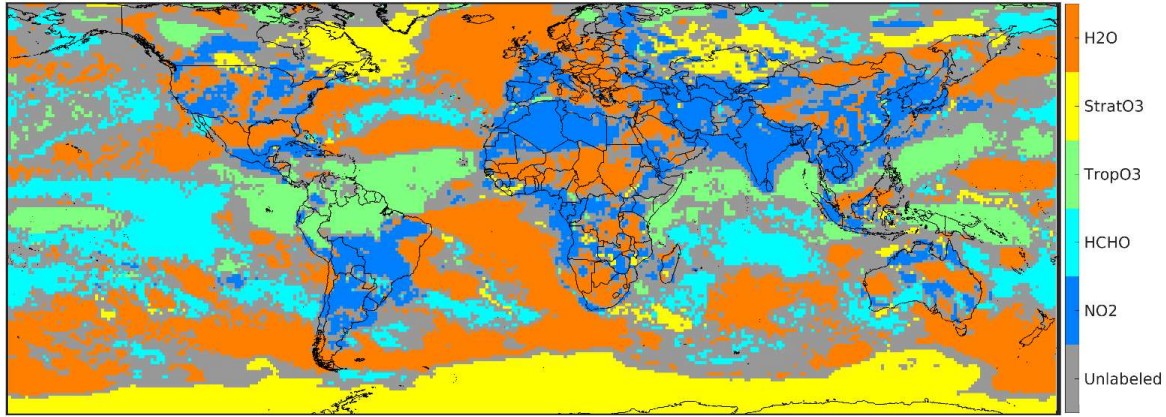

**Figure 7.** The major contributor to TOH trends based on the largest absolute trends of TOH drivers/proxies above 30% of the second most dominant factor.

*3.3.2. Magnitudes of linear trends of TOH key inputs*

Figure 8 shows the linear TOH trends influenced by $NO_2$ (*SOHnitro*), HCHO (*SOHform*), water vapor (*SOHwv*), tropospheric ozone (*SOHtropozone*), and stratospheric ozone (*SOHstratozone*). A discussion on each parameter will follow:



*SOHnitro* – The trends in TOH driven by $NO_2$ show a strong correlation with the a posteriori trend
discussed in Section 3.1.1, with low- and medium-income countries experiencing an increase in TOH due to
rising $NO_2$ levels, while high-income countries see a reduction in TOH due to the opposite trend. The most
significant increase in TOH is observed over India, where both the $NO_2$ trend and TOH sensitivity to $NO_2$ are
prominent. The most rapid regional decline in TOH seems to be over the NCP, because of $NO_x$ reductions that
began after 2011. This finding is particularly noteworthy since M2GMI did not reproduce this trend without
OMI as a constraint. The trend in TOH resulting from $NO_2$ is predominantly anthropogenic in nature. This
aligns with the findings of Chua et al. (2023), who observed that the impact of lightning $NO_x$ emissions on
TOH trends was relatively minor. The global trend in TOH driven by $NO_2$ is positive, but with considerable
variation due to the significant disparities in how anthropogenic $NO_x$ emissions have changed.
*SOHform* – We saw that HCHO was a reasonable proxy for TOH over oceans. Accordingly, the TOH
trends primarily are observed over oceans, especially over the Pacific and the Indian Oceans. This lines up
with the information gathered from the analysis of M2GMI and OMI HCHO observations (Figure 4). These
upward HCHO trends, as discussed in Section 3.1.2, may be influenced by transport and dynamics. It is worth
noting that the increase in TOH tied to this proxy (HCHO) is a global tendency, attributable to the relatively
uniform rise in HCHO levels across oceans.
*SOHwv* –Water vapor is a primary source of OH. The offline sensitivity of ECCOH captures this
tendency (Figure S22). Accordingly, the TOH linear trends mirror those of IWV (Figure S8) with major
increases over oceans. Similar to other experiments, the global TOH increases because of rising water vapor
in the atmosphere. We acknowledge that understanding the reasons for changes in water vapor, which our
model shows to agree with Broger et al. (2022), is a complex subject that goes beyond the scope of our research.
It requires an in-depth understanding of the water cycle, evapotranspiration and precipitation rates, and the
effect of temperature on the air's capacity to hold moisture, known as the Clausius Clapeyron relationship.
However, a great deal of effort has been made to demonstrate that global water vapor levels have increased
significantly in recent decades. This is based on reanalysis data, microwave satellites, and in-situ
measurements (Trenberth et al., 2005; Chen and Liu, 2016; Wang and Liu, 2020; Allan et al., 2023), which is
consistent with what our model shows, as it is well-constrained by MERRA2 reanalysis data.
*SOHtropozone* – The impact of tropospheric ozone on OH formation is widely acknowledged
(Lelieveld et al., 2016). Likewise, our ECCOH offline sensitivity tests have revealed a largely positive
correlation between tropospheric ozone and OH (Figure S23). Consequently, the linear trends observed in
TOH closely mirror those of tropospheric ozone in M2GMI (Figure S3). This tendency is especially noticeable
in the Atlantic Ocean, East and Southeast Asia, as well as the northern region of the Pacific Ocean, where
rising ozone levels have increased TOH. M2GMI suggests that tropospheric ozone levels in the southern
hemisphere have decreased, potentially leading to a downward trend in TOH, an observation that has yet to be
fully confirmed (e.g., Thompson et al., 2021). This finding is especially important given past research
indicating that models tend to exaggerate TOH asymmetry between the northern-southern hemispheres (Strode
et al., 2015; Naik et al., 2013). The decrease in the simulated tropospheric ozone may offer a plausible
explanation for this tendency, but further verification is deemed necessary. Like the previous experiments,
tropospheric ozone on average leads to a global increase in TOH in 2005-2019.
*SOHstratozone* –Stratospheric ozone columns reduce UV actinic fluxes leading to a reduction in
tropospheric $JO^1D$ and thus OH, a tendency well reproduced by ECCOH (Figure S24). Nonetheless,
stratospheric columns did not change noticeably in the tropics and mid-latitudes where OH production is
important; consequently, the linear trends are close to zero or faintly negative due to a slight upward trend in
the column. This tendency results in a rather uniform decrease of TOH globally.



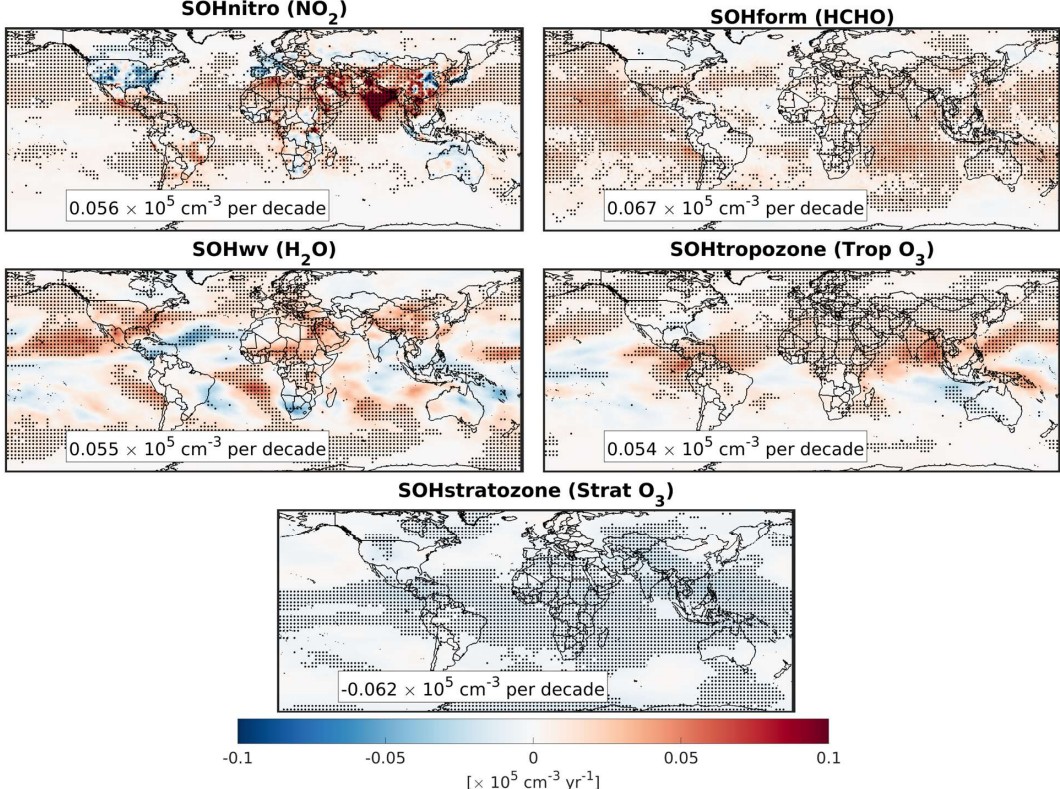

**Figure 8.** The contribution of each TOH key input (addressed in this study) to TOH in 2005-2019. HCHO, $NO_2$, and water vapor results are observationally constrained. Stratospheric ozone columns yielded comparable results compared to total ozone columns observed by OMI, however a large portion of tropospheric ozone trend has remained unverified in the southern hemisphere. ENSO affects the variability of TOH (Anderson et al., 2021), so we add a linear term to Eq.4 that is a function of the Niño 3.4 Index. This helps prevent ENSO from affecting the subsequent results.

### 3.3.3. OMI contributions to TOH trends

It is attractive to gauge the additional information gained from OMI on better representing the linear trends of TOH. To achieve this, we need to analyze three sets of model output: one with OMI scaling factors, one without OMI scaling factors, and one with the $NO_2$ and HCHO drivers (i.e., *SOHnitro* and *SOHform*). The linear trends from these sets of model results are shown in Figure 9. The trends in the first column illustrate the overall effect of $NO_2$ and HCHO on TOH trends, while the two other subplots isolate the effect of OMI from the prior information based on M2GMI. It is immediately apparent that the trends in the driver can be well approximated as the linear combination of the other two experiments, suggesting that the second (or higher) order chemical feedback does not heavily affect the results. M2GMI plays a significant role in shaping the trends in *SOHnitro*, possibly due to the small discrepancy between the trends in OMI and M2GMI columns over regions where TOH is responsive to the driver. The most significant impact of OMI on $NO_2$ is visible over NCP. Concerning HCHO, OMI slows down the upward trends in TOH over oceans which was suggested by M2GMI. In general, M2GMI largely dictates the overall shape of TOH trends driven by $NO_2$ and HCHO possibly due to small difference between the model and OMI observations and/or limited informational content in OMI.



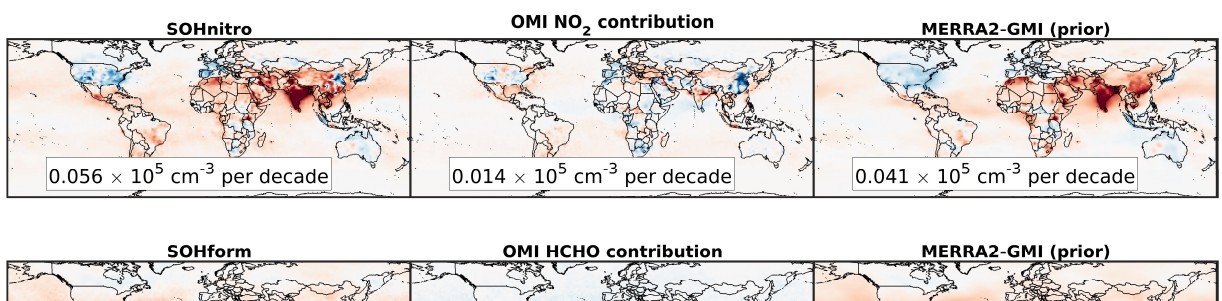

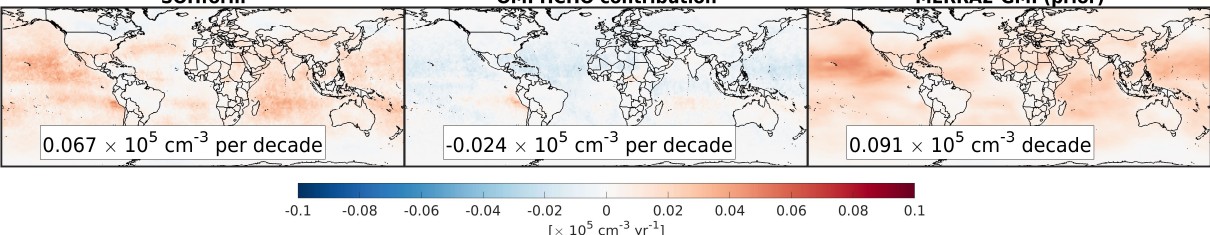

**Figure 9.** The resulting effect of tropospheric $NO_2$ and HCHO on TOH linear trends during 2005-2019 (first column); the contributions of OMI information added on top of the prior knowledge (M2GMI) (middle column); the effect of the prior knowledge on shaping TOH linear trends (last column).

*3.3.4. How well can these experiments explain the simulated trends collectively?*

We find that there is a good degree of correlation between the sum of the linear trends and those of *Sanalysis* ($R^2$=0.65) indicating that a good portion of variability in TOH trend can be well explained by these experiments (Figure S25). Figure 10a shows the linear trend of TOH from *Sanalysis* in 2005-2019, and Figure 10b shows the sum of the linear trends of the five OH key inputs. These maps are one of the most recent and detailed TOH trends available, relative to newer studies (Nicely et al., 2018; Zhao et al., 2020; Chua et al., 2023). The TOH trend from *Sanalysis* varies greatly, where positive values are prevalent over northern parts of the Pacific Ocean, the Middle East, central Africa, and several regions over East Asia. Negative trends are found over the US, southeast Asia, and the southern part of the Pacific Ocean. The linear sum of the experiments strongly aligns with *Sanalysis*, particularly over the northern hemisphere, reinforcing that the selected parameters are sensible choices to reproduce a large portion of variance in TOH trend.

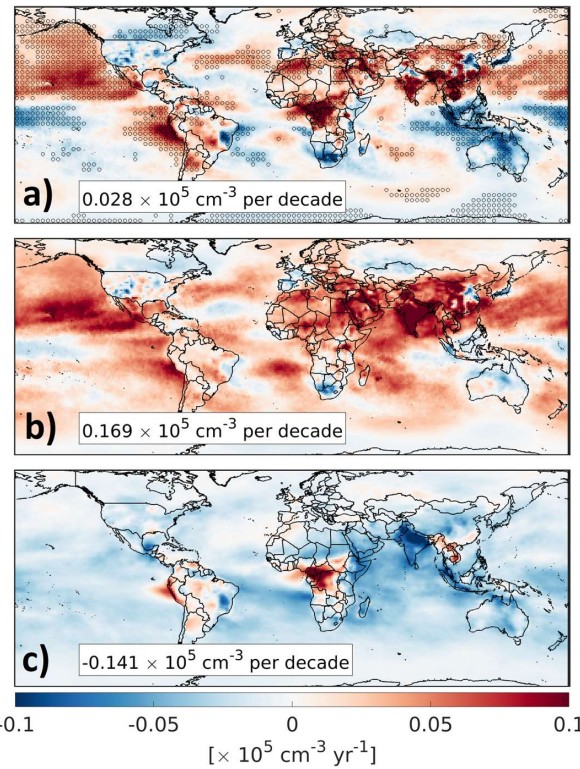

**Figure 10.** (a) The linear trends derived from *Sanalysis* experiment, the "best effort" to simulate the evolution
of the CH₄-CO-OH cycle, from 2005-2019. The statistically significant trends are superimposed by dots. (b)
The linear summation of the five selected TOH influencers including water vapor, NO₂, HCHO, stratospheric
and tropospheric ozone, showing a strong degree of correspondence to the top panel, particularly in the
northern hemisphere. (c) The unexplained portion of the TOH trends, which was not explainable by five
experiments addressed in this research.
Revealing the unexplained portion of TOH trends, which cannot be attributed to the selected TOH
experiments, is necessary. Within the model, various physiochemical factors such as CO, CH₄, dynamics,
aerosols, and clouds can impact the TOH trends. Although we will not delve into these drivers in this study,
we can identify unexplained parts of TOH trends by subtracting the sum of trends derived from the five primary
TOH key inputs from those of *Sanalysis*, which discounts second (or higher) chemical feedback. Figure 10c
displays the unexplained TOH trends between 2005 and 2019. It is readily apparent that there are uniform and
significant downward trends in TOH in the tropics and subtropics where photochemistry is strong. This is most
likely triggered by increasing concentrations of CH₄, which is demonstrated in Figure S10, causing OH levels
to decrease over time. It is very probable that the extent of these downward trends in TOH has been exaggerated
in our model because of the simulated CH₄ increasing too rapidly compared to in-situ observations.
Consequently, the globally-averaged TOH trend derived from *Sanalysis* may be slower than it should be.
Lastly, an unexplained strong upward trend in TOH over central Africa lingers.



## 4. Conclusion


While a comprehensive multi-sensor/multi-species data assimilation and inverse modeling approach,
such as Souri et al. (2020a), Miyazaki et al. (2020), and Souri et al. (2021), would be ideal for fully harnessing
the potential of satellite information on improving multiple aspects of a model representing OH, it will be
prohibitively expensive. Therefore, our simplified approach serves the purpose of understanding the first-order
effects of observational adjustments to TOH drivers/proxies before committing substantial resources to the
implementation/execution of an observationally-constrained, full-chemistry model. Here, we implemented the
newest version of the parameterization of OH, following Anderson et al. (2022), within NASA's GEOS model,
presenting an opportunity to understand and mitigate TOH biases caused by misrepresentation of HCHO and
$NO_2$ concentrations with respect to the state-of-the-art OMI $NO_2$ and HCHO retrievals using Bayesian data
fusion, as well as to unravel the intricacies of TOH to its key inputs such as tropospheric and stratospheric
ozone and water vapor.

We found large positive biases in tropospheric $NO_2$ columns in M2GMI, the archived model used as
an input to the parameterization of OH, compared to OMI over Africa, Eastern Europe, and the Midwest US.
Because of a large positive effect of $NO_2$ (a surrogate for NOx) on TOH, a tendency well captured by our
implicit parameterization, these overestimations introduced significant regional biases in TOH up to 20%, and
a global overestimation of TOH by 1%. Consistent with former work, we saw distinct disparities in the sign
of linear trends of tropospheric $NO_2$ over high- and medium-income countries (i.e., negative) and low-income
countries (i.e., positive). While M2GMI generally replicated these trends, notable deviations were identified
over China leading to an erroneous trend of TOH.

Pronounced inaccuracies with regards to both the simulated HCHO magnitude and trend in M2GMI
were revealed by OMI over land. However, this proxy for OH was loosely connected to TOH in areas where
photolysis was not the major sink of HCHO (Wolfe et al., 2019), especially over land. Over oceans, where
HCHO and TOH were highly correlated, adjustments to M2GMI by OMI HCHO were relatively mild resulting
in small alterations to TOH which was by an order of magnitude lower than those of $NO_2$. These mild
alterations speak to either an insufficient amount of information in OMI or the reasonable accuracy of M2GMI
over pristine areas.

In general, five variables including $NO_2$, HCHO, water vapor, tropospheric ozone, and stratospheric
ozone, could collectively account for 65% of the variance in TOH trends globally. To estimate this, we
executed various modeling experiments to isolate the effect of $NO_2$, HCHO, water vapor, tropospheric ozone,
and stratospheric ozone on long-term trends of TOH in 2005-2019 at 1°×1° resolution. Except for tropospheric
ozone, these variables were either constrained by observations or aligned with independent observations,
boosting confidence in our trend results. Given the robust positive correlation between OH and $NO_2$, HCHO,
water vapor, and tropospheric ozone over regions where photochemistry was active, TOH trends influenced
by these variables closely mirrored the trends in their respective drivers/proxies. For instance, high- and
medium-income countries exhibited negative TOH trends driven by $NO_2$. Rising tropospheric ozone over east
and south Asia, heavily vetted by various observations (Guadel et al., 2018), led to an upswing in TOH over
the Pacific Ocean. The trend of water vapor, greatly in agreement with independent observation (Broger et al.,
2022), was dominantly positive over oceans leading to further enhancement of TOH. Rising HCHO over
Pacific and Indian Ocean suggested by constrained M2GMI was associated with increased TOH. The effect
of stratospheric ozone on TOH was marginal in low and mid latitudes due to negligible changes in stratospheric
ozone columns in M2GMI reconfirmed by OMI total ozone column observations.

A large offset between our analysis experiment with varying CO and $CH_4$ concentrations was observed
after removing the sum of the linear trends derived from these five key experiments from the analysis
experiment, indicating that our future research using ECCOH should include new experiments isolating the
effects of CO, $CH_4$, and transport (e.g., Gaubert et al., 2017; Zhao et al., 2020). Those experiments will refine
the investigation of the unexplained portion of the TOH trend.

The development of an effective parameterization of OH, that is capable of integrating advanced
satellite-based gas retrievals and improved weather forecast models enabled us to unravel the convoluted
response of TOH to various parameters. Nonetheless, it is important to recognize some of the limitations



associated with our work: the offline nature of the Bayesian data fusion algorithm makes the entire experiment
blind to the interconnected responses of various compounds, such as ozone or aerosols, to adjustments to $NO_2$
and HCHO. Despite this limitation, our work has provided valuable insights into the first-order effects of
adjustments on TOH key inputs. This can help quickly identify areas where our prior knowledge is least
reliable to simulate TOH.
The longevity and stability of Aura's record of observations have played a significant role in
constraining/assessing several important variables pertaining to TOH on a global scale. This is exemplified by
the wealth of information obtained from OMI $NO_2$, HCHO, water vapor, total ozone columns, and Microwave
Limb Sounding (MLS) temperature and ozone, that are used directly or indirectly in our analysis. However,
as Aura's mission comes to an end, there will be a gap in the monitoring of these variables. TROPOMI, OMI's
successor, can help fill this gap, but its record of observation is still short; therefore, it is important to invest
in research to harmonize data from multiple satellite observations such as OMI and TROPOMI (e.g., Hilboll
et al., 2013). This is because each sensor can have different biases and spatial representativity, which can lead
to inconsistencies and potentially conflicting values if they are used together.

## Acknowledgements

This research was supported by the National Aeronautics and Space Administration (NASA) Aura Mission
project science funds. We thank Gonzalo Gonzalez Abad for sharing OMI HCHO v4 data.

## Data availability

Satellite data can be accessed for Level 2 OMI tropospheric $NO_2$ at
https://doi.org/10.5067/MEASURES/MINDS/DATA204 (Lamsal et al., 2022), Level 2 OMI total ozone
columns at https://disc.gsfc.nasa.gov/datasets/OMTO3_003/summary (Bhartia, 2005), OMI SAO HCHO at
https://waps.cfa.harvard.edu/sao_atmos/data/omi_hcho/OMI-HCHO-L2/ (Gonzalez Abad, 2023), MOPITT
CO (https://doi.org/10.5067/TERRA/MOPITT/MOP03JM_L3.008) (NASA LARC, 2000), OMI/MLS $TO_3$
at https://acd-ext.gsfc.nasa.gov/Data_services/cloud_slice/data/tco_omimls.nc (Ziemke, 2023).
In-situ CO and $CH_4$ observations can be obtained from
https://gml.noaa.gov/dv/data/index.php?category=Greenhouse%2BGases (Helmig et al., 2021; Lan et
al.,2021).
MERRA2-GMI model outputs can be downloaded from https://acd-
ext.gsfc.nasa.gov/Projects/GEOSCCM/MERRA2GMI/ (NASA Goddard Space Flight Center, 2023).

## Code availability

OI-SAT-GMI python package developed for this research can be found from
https://doi.org/10.5281/zenodo.10520136 (Souri, 2024).
GEOS-Quickchem used to run the modeling experiments encompassing ECCOH can be found from
https://github.com/GEOS-ESM/QuickChem.git.
GEOS model can be obtained from https://github.com/GEOS-ESM/GEOSgcm.git.
Offline ECCOH calculations to derive the sensitivity of TOH to different drivers/proxies can be obtained
from https://doi.org/10.5281/zenodo.10685100

## Authors contributions

A.H.S and B.N.D designed the research. A.H.S analyzed the data, conducted the simulations, made all the
figures, and wrote the original manuscript. B.N.D helped with conceptualization, fund raising, and writing.
S.A.S helped configuring the model and interpreting the results. M.E.M and D.C.A implemented the improved
ECCOH module into GEOS-5 Quickchem. J.L. thoroughly validated the model with respect to CO and $CH_4$
observations. B.W. provided an improved CO emission inventory. L.D.O provided M2GMI and helped
interpret it. Z.Z. provided improved wetland $CH_4$ emissions. All the authors contributed to the discussion and
edited the paper.

## Competing interests

B.N.D is a member of the editorial board of Atmospheric Chemistry and Physics.



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
