# Peer review of "Enhancing Long-Term Trend Simulation of Global 3 Tropospheric OH and Its Drivers from 2005-2019: A 4 Synergistic Integration of Model Simulations and Satellite 5 Observations"

_EGUsphere, 2024_

## Referee Comment (RC1)

General comments:

The manuscript "Enhancing Long-Term Trend Simulation of Global Tropospheric OH and Its Drivers from 2005-2019: A Synergistic Integration of Model Simulations and Satellite Observations" estimates the tropospheric OH trend lead by $NO_2$, tropospheric ozone, $H_2O$, and HCHO on 1x1 model resolution. Quantifying the drivers of the global OH changes is essential for better understanding recent changes in the global $CH_4$ mixing ratio. The manuscript is generally well organized and clearly written. In this study, the ECCOH model estimates OH by machine learning method. Machine learning predicts OH by finding the correlation patterns between OH and the input factors. The cause-and-effect relationship is not necessarily captured by the machine learning method. The authors estimated the drivers of the OH trend based on the sensitivity of OH to different factors as given by machine learning parameterized OH. My main concern is whether the sensitivity of OH to different factors estimated by machine learning parameterization is consistent with that simulated by the M2GMI model. Is there any possibility of evaluating the sensitivity? Besides this, I only have a few minor comments. I recommend the paper be published on ACP after addressing these comments.

Specific Comments:

1 L25: How is the TOH estimated? Is it weighted by air mass, volume, or $CH_4$ reaction?

2 L191-193: Are the VOCs simulated by M2GMI distributed only in the first layer of the model? Why was the CO produced by VOCs released to the first vertical level of the model?

3 L224 "**E** is populated by the average sum of precision error squares the satellite product provides" . "**E**" should include instrument, representation, and forward model errors. However, here only the instrument error is included.

4 L225-226: The "mass-conserved linear barycentric interpolation method" should be described here.

5 L248: In my understanding, the chemical compounds including tropospheric ozone are prescribed in the ECCOH model. How do the improved $NO_2$ and HCHO represent for more accurate simulation of other chemical compounds?

6 Equation 4 what is the temporal resolution of y? When $\omega = 1$, the cosine function has a period of 1, how does it account for the seasonal cycle?

7 L265: How to use the Levenberg–Marquardt algorithm to optimize the estimation?

8 L359-L363: It is confusing here, do you mean the water vapor in the "Sanalysis" experiment the water vapor is from the GOES online simulation while in the SOHvv simulation, the water vapor is from the MERRA2 reanalysis?

9 L412-416: The Bayesian system gives low AK over the remote areas because the satellite observations give higher relative error over the regions with low $NO_2$ values while the B is arbitrarily set to 50% for all the model grid. Considering the $NO_2$ simulated by M2GMI may also have larger relative uncertainties over the remote areas, "low AK in remote areas shows rich information from OMI tropospheric $NO_2$ gravitates more polluted regions. " is not a robust conclusion.

10 Figures S1 and S2, Are the grey regions in the figures indicating a non-significant trend? It seems that the M2GMI failed to capture the positive trend over most of the positive trends in tropospheric ozone over the Northern hemisphere, and over the tropical ocean, the M2GMI simulated a significant negative trend, which is not observed by the OMI/MLS data.

11 L529-543: Here is my main concern for this paper. Although the machine learning

approach can reproduce the OH distribution, how well the machine learning method can reproduce the sensitivity of OH to $NO_2$, HCHO, tropospheric $O_3$, etc. is not evaluated in Anderson et al. (2022; 2023). Nice et al. (2018) estimated that the $NO_x$ increase can lead to a decrease in OH concentrations over the high $NO_x$ regions. The negative sensitivity accounts for 10% of all the cases tested by the chemical box model. As shown in Figure 5, machine learning gives overall positive sensitivity. Also, for HCHO, which acts as both OH sink and $HO_2$ source, machine learning gives overall positive sensitivity. The sensitivity calculated by machine learning can have a large impact on the conclusion of this study. Is there any possibility to evaluate the sensitivity estimated by machine learning?

12 L543-543; L731: Are the increase in $CH_4$ means that the model is not fully-spin-up? Usually, 3 times of lifetime is required to reach a steady state.

13 L717-723: Does the global reduction of CO emissions contribute to the unexplained TOH trend?

---

## Author Comment (AC1)

General comments:

The manuscript "Enhancing Long-Term Trend Simulation of Global Tropospheric OH and Its Drivers from 2005-2019: A Synergistic Integration of Model Simulations and Satellite Observations" estimates the tropospheric OH trend lead by NO2, tropospheric ozone, H2O, and HCHO on 1x1 model resolution. Quantifying the drivers of the global OH changes is essential for better understanding recent changes in the global CH4 mixing ratio. The manuscript is generally well organized and clearly written. In this study, the ECCOH model estimates OH by machine learning method. Machine learning predicts OH by finding the correlation patterns between OH and the input factors. The cause-and-effect relationship is not necessarily captured by the machine learning method. The authors estimated the drivers of the OH trend based on the sensitivity of OH to different factors as given by machine learning parameterized OH. My main concern is whether the sensitivity of OH to different factors estimated by machine learning parameterization is consistent with that simulated by the M2GMI model. Is there any possibility of evaluating the sensitivity? Besides this, I only have a few minor comments. I recommend the paper be published on ACP after addressing these comments.

| Response |
| --- |
| **We thank the reviewer for their constructive comments and the major point they raised about the evaluation of the response of OH to several parameters. We would like to address this comment in three ways:** *the nature of this statistical approach vs. an explicit full chemistry model*, *qualitative agreement of the tendency of the sensitivity with theoretical expectations*, **and** *providing evidence of the capability of the framework at picking up non-linearities*.

 **The nature of this statistical approach vs. an explicit full chemistry model:**

 **The limitation of statistical approaches in fully resolving "cause and effect" has been widely recognized in literature. Similarly, in the case of the ECCOH, we cannot entirely disentangle causation from correlation based solely on establishing a relationship between the distribution of OH and its influencers. As discussed in the paper and related references, the aim of the ECCOH framework is not to replicate all physical capabilities of a physics-based model, but rather to provide a first-order sensitivity experiment capable of qualitatively studying the effect of the biases in the input data on OH. This allows for more systematic strategies for full chemistry runs.**

 **Robust evaluation of the sensitivities are mainly limited by the implicit nature of statistical methods. The implicit nature of statistical methods makes it difficult to identify a specific physiochemical process in a full chemistry model that is representative of the perturbation made to the OH parameterization. For example, perturbation of NO$_2$ in ECCOH represents the empirical relationship between OH and NO, which involves multiple processes, including NO+HO$_2$ (RO$_2$), NO$_2$+OH, the formation of ozone, aerosol HOx update, and radiation; thus, it is not restricted to specific chemical reaction. However, we do not know to what extent these processes are implicitly included in the perturbations. Similarly, the perturbation of HCHO does not necessarily represent the HCHO+OH reaction but rather where this compound can be used as a proxy for OH (discussed later).**

 **Despite these challenges, it is important to recognize that the aim of ECCOH was not uniquely to improve the estimation of OH. Instead, the parameterization includes various input parameters (~ 27 inputs) so that the machine-learning algorithm could better understand the relationship between OH and its drivers for a wide range of atmospheric conditions. Because of this reason ECCOH has been able to represent OH distributions for extreme events (such as El Nino) that were not used during the training (Anderson et al., 2022, 2023, 2024). Without a proper** |

establishment of the sensitivities (i.e., right OH prediction for a wrong reason), we would not have been able to reproduce OH distributions for such events.

ECCOH exhibits greater flexibility than some statistical approaches that use simpler assumptions (Valin et al., 2016; Murray et al., 2021; Wolfe et al., 2019; Pimlot et al., 2022), which may not fully capture complexities associated with the real atmosphere such as the effect of clouds and surface albedo or be applicable over high HOx conditions. For example, the global approximation of OH as function of reactive nitrogen formulated in Murray et al. (2021) may not be applicable for a wide range of atmospheric conditions. Anderson et al. (2022) carefully selected a vast number of OH-related parameters allowing us to better represent the response of OH to its drivers over both land and ocean and should be considered an improvement towards enhancing statistical-based OH studies.

**The sensitivities qualitatively agree with our theoretical expectations:**

$NO_2$ – It is believed to have positive feedback of reactive nitrogen on tropospheric OH based on both physics-based and statistical studies (Zhao et al., 2019, 2020, He et al. 2021, Chau et al., 2023; Naik et al., 2013; Murray et al., 2013; Strode et al., 2015; Nicely et al., 2018). Likewise, our perturbations in $NO_2$, as a surrogate for reactive nitrogen, causes TOH to increase. This increase happens in the tropospheric region. We will show in this response letter that non-linearities occur when we do the perturbations only at the surface layer where $NO_x$ is elevated because of different reactions such as ozone titration and $NO_2+OH$ which can reduce OH.

HCHO –We used this compound as a proxy (and not a driver) of OH following the studies of Valin et al. (2016) and Wolfe et al. (2019). The equation provided by Wolfe et al. (2019) follows:

$$[HCHO] = \frac{\alpha k'_{OH}[OH] + P_0}{j_{HCHO} + k_{HCHO+OH}[OH]}.$$

where the numerator is the production of HCHO from the oxidation of background VOCs, and the denominator is the sink of HCHO through both photolysis and the reaction with OH. In remote regions where $j_{HCHO} >> k_{HCHO+OH}[OH]$, we can safely ignore the reaction of HCHO+OH, and assuming the minor source ($P_0$) to be zero, [HCHO] and [OH] become linearly correlated suggesting that if see a higher amount of HCHO, there has been more OH to oxidize background VOCs, assuming the slope stays constant (varies only by 5% based on Wolfe et al. (2016)). In high HOx regions, [HCHO] and [OH] becomes decoupled. This is what we essentially see from the perturbations in HCHO meaning the ECCOH response coincides with the theoretical expectation.

Stratospheric ozone – More stratospheric ozone hampers actinic flux leading to less production of $jO^1D$ resulting in a negative relationship between this quantity and OH.

Tropospheric ozone and water vapor – Both are primary source of OH and show positive feedback on OH. The magnitude of the positive sign can be influenced by the underlying surface albedo, clouds, or other implicit processes.

Now, the pivotal question is can we quantitatively assess these numbers? Since the perturbation of OH drivers in ECCOH are a snapshot of perturbing one variable without considering the response of unperturbed ones, we think it is challenging to replicate the identical experiments in a full chemistry model. Besides, the implicit nature of ECCOH makes it difficult to know exactly

which physiochemical processes we should collectively pick from a full chemistry model to compare with. Therefore, our confidence in perturbation has been mostly achieved through the "weight of evidence". As a result, all experiments done in our draft should be seen from a statistical and somewhat qualitative perspective.

_**Providing evidence of the capability of the framework at picking up non-linearities**_

This reviewer raised a valid concern about the perturbation of $NO_2$. They pointed out that we did not have any negative values, but it is expected to see negative tendencies over extremely polluted regions where $NO_x$ can hamper OH levels. There are two reasons behind this. First, we focused on the tropospheric region where the majority of vertical grid boxes do not experience elevated $NO_x$ levels. Second, the M2GMI grid resolution is not spatially fine enough to fully resolve non-linear chemistry. However, this is not a major concern for the methane-CO-OH studies, as we intend to use ECCOH for climate studies at coarse resolution and not urban air quality applications. To demonstrate the capability of ECCOH at capturing negative sensitivities for more polluted regions, we applied the perturbation at the surface and calculated the changes in the surface OH mixing ratio. We indeed saw large negative values over polluted regions (shown later), which would be expected from a combined effect of ozone titration and $NO_2+OH$ in a full chemistry model.

The capability of XGBoost at solving non-linear tendencies has been proven extensively (e.g., Johnson and Zhang, 2014: https://arxiv.org/pdf/1109.0887).

**Modifications**

To account for the reviewer's comment, we added more caveats throughout the paper and also included the perturbation experiments related to surface $NO_2$.

In the abstract, the description of the results is already qualitative. But to clarify, we added: "This innovative module helps efficiently predict the convoluted response of TOH to its drivers/proxies in a statistical way."

In section 2.2.3, right after introducing the method to get the perturbations:

"It is crucial to acknowledge that ECCOH has established an implicit relationship between OH and various input parameters statistically. These perturbations could involve a range of physiochemical processes that are challenging to fully decipher. For example, the perturbation of $NO_2$, acting as a surrogate of reactive nitrogen, involves chemical reactions that include reactive nitrogen like $NO+HO_2$ and $NO_2+OH$, ozone formation, aerosol $HO_x$ uptake, and radiation. Nonetheless, it may not be feasible to understand to what extent these processes have been represented by ECCOH. Therefore, the presented perturbations in this work should be viewed qualitatively."

In the results and discussion:

"Deciphering the precise chemical processes influencing the response of OH to $NO_2$ using a machine-learning approach is challenging. However, it is widely recognized that reactive nitrogen has positive feedback on tropospheric OH through increased $NO+HO_2$ and ozone (Murray et al., 2021; Zhao et al., 2020; He et al., 2021). Considering $NO_2$ as a surrogate for reactive nitrogen, similar tendencies are expected, as evident from the positive numbers from the sensitivity results obtained from offline calculations. The response of TOH to $NO_2$ displays a pronounced seasonal cycle stemming mainly from

photochemistry. It is believed to have some negative values for the sensitivity of OH to $NO_2$ for extremely polluted regions due to radical termination through $NO_2+OH$ or ozone titration (Nicely et al., 2018). While we have not identified any negative values in the tropospheric domain, we have observed significant negative values of OH when perturbing $NO_2$ at the model surface layer (Figure S*). This tendency highlights the ECCOH's ability to account for non-linearities."

[Figure]

**Figure S\*.** The sensitivity of surface OH to $NO_2$ perturbations in offline ECCOH in four different seasons.

**To emphasize the relationship between HCHO and OH, we copy our discussion regarding HCHO response map here:** "*The interplay between HCHO and OH is contingent on the intricate dynamics governing HCHO production from the oxidation of VOCs and methane and HCHO loss from various chemical pathways (Valin et al., 2016; Wolfe et al., 2019). In remote areas where $HO_x$ is low, the prevailing sink of HCHO is through photolysis. Conversely, in more polluted areas, the reaction of HCHO+OH emerges as a competing loss pathway. Assuming a steady-state approximation, which is a reasonable assumption for pristine areas, the photolysis loss of HCHO dominates over the reaction with OH, resulting in a linear relationship between HCHO and OH. In other words, high (low) HCHO concentrations are indicative of high (low) TOH. It is because of this that we use HCHO as a proxy of TOH in remote oceans regions. In regions characterized by heightened $HO_x$ levels, OH and HCHO become decoupled. Encouragingly, our implicit parametrization of OH has considerable skill at elucidating these intricate chemical tendencies; specifically, it reveals muted responses in regions with relatively tangible pollution levels, whereas positive responses are evident in oceanic regions. Like results obtained for $NO_2$, the response map has a seasonal cycle due to photochemistry.*" **The qualitative description of the response map for other factors had been provided in section 3.3.2.**

**In the conclusion section:**

"The development of an effective parameterization of OH, that is capable of integrating advanced satellite-based gas retrievals and improved weather forecast models enabled us to unravel the convoluted response of TOH to various parameters. Nonetheless, it is important to recognize some of the limitations associated with our work: first, the offline nature of the Bayesian data fusion algorithm makes the entire experiment blind to the interconnected responses of various compounds, such as ozone or aerosols, to adjustments to $NO_2$ and HCHO. Despite this limitation, our work has provided valuable

insights into the first-order effects of adjustments on TOH key inputs. This can help quickly identify areas where our prior knowledge is least reliable to simulate TOH. Second, the machine learning algorithm employed for parameterizing OH is implicit and its response to drivers/proxies is complex, making it difficult to quantitatively verify against full chemistry models. However, by including a vast number of parameters in the parameterization, Anderson et al. (2022) boosted its ability to understand the convoluted chemistry of OH. This has allowed for reproducing OH for events not included in the training dataset (Anderson et al., 2022, 2023, 2024)."

Specific Comments :

L25: How is the TOH estimated? Is it weighted by air mass, volume, or CH4 reaction?

| Response |
| --- |
| **Throughout the paper, TOH is weighted by the CH$_4$ reaction following Lawrence et al. (2001).** |

| Modifications |
| --- |
| **To better highlight this, we have moved the sentence describing this way of estimation to the ECCOH description part (Section 2.1.2):**
 "*Throughout the paper, TOH is determined based on the methane-reaction-weighted OH suggested by Lawrence et al. (2001).*" |

L191-193: Are the VOCs simulated by M2GMI distributed only in the first layer of the model? Why was the CO produced by VOCs released to the first vertical level of the model?

| Response |
| --- |
| **No, M2GMI simulates the 3D distribution of VOCs. Because ECCOH is not a full chemistry mechanism (O3-NOx-VOC-Halogen-Aerosol) and a non-negligible fraction of CO is produced through VOC oxidation, we needed to "approximate" these secondary-formed contributions based on yield estimations from Duncan et al. (2007). These effective yields are co-emitted at the same location where CO is emitted, which is primarily at the surface. Our model evaluation against MOPITT CO profiles, columns, and NOAA's measurements showed reasonable results despite this simplified approximation. (Figures S\*).**

 **We had tested an alternative approach (https://github.com/ahsouri/OI-SAT-GMI/blob/main/tools/create_ind_CO_emiss.py) by which we generate CO production rates based on summing important reactions rates yielding CO from M2GMI in "4D", so that we would avoid placing secondary-produced CO emissions at the surface. Although this method in theory is more realistic, we noticed large biases in our model compared to observations such as MOPITT. These unwanted biases were caused by large negative biases of M2GMI CO simulations. So, we resort to taking the more simplified approach.** |

L224 "E is populated by the average sum of precision error squares the satellite product provides" . "E" should include instrument, representation, and forward model errors. However, here only the instrument error is included.

| Response |
| --- |
| **We agree with the reviewer that in a fully-cycled data assimilation or an inversion framework, this term should encompass both the forward model and the representation errors. Nonetheless, our approach, although adopted from Bayesian framework, is a data fusion algorithm and the** |

forward model operator is not necessary (= 1.0). Moreover, because we upscaled OMI grids to M2GMI to have both at the coarsest resolution grid, the spatial representation errors related to unresolved scales in M2GMI relative to OMI footprint have been considered using the mass-conserved barycentric interpolation method. Admittedly, the unresolved processes in M2GMI have been overlooked. So we added a caveat.

| Modifications |
|---|
| **We added:**
"**E** is the sum squares of error covariance matrix of the observations and the representation errors, **Y** is the observations, and **H** is the observational operator which is equivalent to the identity matrix in our case. The instrument error part of **E** is populated by the average sum of precision error squares the satellite product provides. We interpolate both **E** and **Y** into the M2GMI grid box using a mass-conserved linear barycentric interpolation method. This interpolation method removes the spatial representation error resulting from the unresolved scales in M2GMI columns. Nonetheless, we did not take into account the errors of unresolved processes in M2GMI to augment to **E**." |

L225-226: The "mass-conserved linear barycentric interpolation method" should be described here.

| Response |
|---|
| **Sure, we elaborated on this method, publicly available at https://github.com/ahsouri/OI-SAT-GMI/blob/main/oisatgmi/interpolator.py** |

| Modifications |
|---|
| **We added:**
"We interpolate both **E** and **Y** into the M2GMI grid box using a mass-conserved linear barycentric interpolation method. In this method, both OMI's observations and errors in the L2 granules provided at their irregular grid have been projected into a common grid of 0.25×0.25 degrees using Delaunay triangulation bi-linear interpolation. Subsequently, we convolve these re-gridded maps with a box filter whose kernel size is equivalent to the rounded fraction of M2GMI grid box size to the re-grided OMI pixel size based on formulation in Souri et al. (2022)." |

L248: In my understanding, the chemical compounds including tropospheric ozone are prescribed in the ECCOH model. How do the improved NO2 and HCHO represent for more accurate simulation of other chemical compounds?

| Response |
|---|
| **That is the major drawback of a data fusion algorithm (a post-processor) over doing an actual inversion+data assimilation (e.g., Souri et al., 2020: https://acp.copernicus.org/articles/20/9837/2020/). Since the goal of ECCOH is to efficiently reproduce CH4-OH-CO only and not full-chemistry, the data fusion algorithm still has the capability to enhance the representation of OH. We have mentioned the limitation in the conclusion and the method description. We are highlighting them here:**

"*In our approach, the adjustments are implemented to the M2GMI output (i.e., a data fusion approach instead of data assimilation one), thereby **restricting** the full use of improved $NO_2$ and HCHO representation for more accurate simulation of other chemical compounds impacted by $NO_2$ and HCHO, including ozone (e.g., Souri et al., 2020a, 2021).*"

**In the conclusion section:** |

> "*Nonetheless, it is important to recognize some of the limitations associated with our work: the offline nature of the Bayesian data fusion algorithm makes the entire experiment blind to the interconnected responses of various compounds, such as ozone or aerosols, to adjustments to NO2 and HCHO. Despite this limitation, our work has provided valuable insights into the first-order effects of adjustments on TOH key inputs. This can help quickly identify areas where our prior knowledge is least reliable to simulate TOH.*"

Equation 4 what is the temporal resolution of y? When $\omega = 1$, the cosine function has a period of 1, how does it account for the seasonal cycle?

| Response |
| --- |
| **The temporal resolution of y is monthly. "t" is in the fractional year minus the first year of data. So, it starts from 0 (=2005) and ends at 15 (=2020). When the cosine function has a period of 1, its maximum is at each year and its minimum is in the middle. Of course, the phase parameter can shift the seasonal cycle phase. The equation (including the phase) is equivalent to summing sin and cosine functions (e.g., https://agupubs.onlinelibrary.wiley.com/doi/full/10.1029/2010GL044245).** |

| Modifications |
| --- |
| **We added:** |
| "The equation comprises several variables, including $y$ (data points) on monthly-basis, $a_0$ as the mean, $a_1$ as the linear trend, $t$ as time (fractional year), $a_{i+1}$, $\omega_i$, and $\varphi_i$ are the amplitude, frequency, and phase, respectively." |

L265: How to use the Levenberg–Marquardt algorithm to optimize the estimation?

| Response |
| --- |
| **This is a common optimizer that combines both Gauss-Newton and gradient descent algorithm for fitting non-linear curves without having to worry about the bounds. It has been widely used in various fields. We have added a reference. We used https://docs.scipy.org/doc/scipy/reference/generated/scipy.optimize.curve_fit.html.** |

| Modifications |
| --- |
| **We added:** |
| "This estimation is optimized using the Levenberg–Marquardt algorithm (Marquardt et al., 1996) using *SciPy* open-source package." |

L359-L363: It is confusing here, do you mean the water vapor in the "Sanalysis" experiment the water vapor is from the GOES online simulation while in the SOHvv simulation, the water vapor is from the MERRA2 reanalysis?

| Response |
| --- |
| **Yes, in SOHwv, the water vapor fields fed to OH paramterization is fixed to MERRA2 monthly-varying 2005 fields. We reworded it.** |

| Modifications |
| --- |

> "Amongst various OH drivers/proxies studied here, water vapor is simulated online based on the GEOS simulation; to conduct *SOHwv* which aims at isolating the water vapor effect on OH without affecting meteorology, we set water vapor fields fed to the parametrization of OH to the offline MERRA2 based on the monthly-varying 2005 simulations. Simultaneously, GEOS is allowed to simulate water vapor online to address meteorology. This ensures that the meteorology remains consistent across both *SOHwv* and *Sanalysis*."

L412-416: The Bayesian system gives low AK over the remote areas because the satellite observations give higher relative error over the regions with low NO2 values while the B is arbitrarily set to 50% for all the model grid. Considering the NO2 simulated by M2GMI may also have larger relative uncertainties over the remote areas, "low AK in remote areas shows rich information from OMI tropospheric NO2 gravitates more polluted regions. " is not a robust conclusion.

| Response |
| --- |
| **We understand the reviewer's concern. We first want to clarify that we only initialized "B" to 50% but it is regularized (inflated) based on all observed pixels. Since most of the pixels covering the globe are in less polluted regions (i.e., 70% of our planet is covered by oceans which tend to have low NOx), the regularization factor (>>1) is dictated by observations in those regions. So, despite inflating B, we can't gain much information from OMI. We observed the same tendency using OMI for inverting NOx in East Asia (Souri et al., 2020). For that study, we set biogenic and biomass burning emissions errors up to 300% and still saw low AK in non-polluted regions (see Figure 3 in https://acp.copernicus.org/articles/20/9837/2020/). However, we agree that these errors should be tackled in a more systematic way using the NMC method so we can have more confident in this conclusion.** |

| Modifications |
| --- |
| **We decided to add a caveat:**

"In other words, it is difficult to have high confidence in the degree of deficiency the model can have in simulating $NO_2$ over pristine areas by comparing it to OMI. This notion mathematically manifests in low AK in remote areas showing that rich information from OMI tropospheric $NO_2$ gravitates more towards polluted regions. This finding assumes that the regularized covariance matrix of the prior error does not substantially vary between land and ocean and is isotropic." |

Figures S1 and S2, Are the grey regions in the figures indicating a non-significant trend? It seems that the M2GMI failed to capture the positive trend over most of the positive trends in tropospheric ozone over the Northern hemisphere, and over the tropical ocean, the M2GMI simulated a significant negative trend, which is not observed by the OMI/MLS data.

| Response |
| --- |
| **Yes, we added in the caption what the gray areas are. The discrepancy between M2GMI and OMI/MLS data in the southern hemisphere has been a topic for debate for several years. Tropospheric ozone trends from various satellites/retrievals do not have consensus on whether the upward trend in the southern hemisphere is realistic or not. We have mentioned that in Text S1:**
"The trends observed in the southern hemisphere by OMI/MLS do not align with those simulated by MERRA2-GMI (Ziemke et al., 2019). Lu et al. (2019) indirectly supports the upward trends detected by OMI/MLS by compiling long-term records (1990-2015) of several surface observations and ozonesonde measurements at high latitudes in the southern hemisphere. Using a global model, they |

hypothesized that the upward trends may resulted from the expansion of Hadley circulation, particularly in austral autumn (March until May), leading to greater stratospheric contributions to the surface and a more effective mixing of ozone precursors from heavily polluted tropical regions to the free troposphere. However, it remains unverified whether this expansion occurs throughout the year, resulting in widespread upward trends observed by OMI/MLS. A more recent study, Thompson et al. (2021), utilized Southern Hemisphere Additional Ozonesondes (SHADOZ) data and suggested that the free-tropospheric ozone trends in 1998-2019 were fainter than those detected by satellite observations and varied greatly from season to season due to atmospheric dynamics (i.e., expansion or shrink in the tropopause height). An important caveat to consider is that satellite-based tropospheric ozone concentration can be largely uncertain due to limited sensitivity of the observed radiance to the optical path of ozone in the lower tropospheric region. Gaudel et al. (2018) have compiled the tropospheric ozone trends observed by different satellites and retrieval algorithms and observed that most of them support the upward trends in Asia. However, there is vast disagreement when it comes to the southern hemisphere. Therefore, there is rather insufficient evidence to support the strong upward trends detected by OMI/MLS in the southern hemisphere, nor can it be claimed that MERRA2-GMI has reproduced reasonable trends in that region."

**And in the discussion:**
"M2GMI suggests that tropospheric ozone levels in the southern hemisphere have decreased, potentially leading a downward trend in TOH, an observation that has yet to be fully confirmed (e.g., Thompson et al., 2021). This finding is especially important given past research indicating that models tend to exaggerate TOH asymmetry between the northern-southern hemispheres (Strode et al., 2015; Naik et al., 2013). The decrease in the simulated tropospheric ozone may offer a plausible explanation for this tendency, but further verification is deemed necessary."

L529-543: Here is my main concern for this paper. Although the machine learning approach can reproduce the OH distribution, how well the machine learning method can reproduce the sensitivity of OH to NO2, HCHO, tropospheric O3, etc. is not evaluated in Anderson et al. (2022; 2023). Nice et al. (2018) estimated that the NOx increase can lead to a decrease in OH concentrations over the high NOx regions. The negative sensitivity accounts for 10% of all the cases tested by the chemical box model. As shown in Figure 5, machine learning gives overall positive sensitivity. Also, for HCHO, which acts as both OH sink and HO2 source, machine learning gives overall positive sensitivity. The sensitivity calculated by machine learning can have a large impact on the conclusion of this study. Is there any possibility to evaluate the sensitivity estimated by machine learning?

**Response**

**Thanks for raising this important aspect. We have addressed this major point at the top.**

L543-543; L731: Are the increase in CH4 means that the model is not fully-spin-up? Usually, 3 times of lifetime is required to reach a steady state.

**Response**

**Although we cannot completely disregard the impact of initial conditions on our results, we started the model ten years prior with initial conditions that were spun up for another decade based on another study from our group. Upon examining Figure S10, we observed that the initial state of our simulations followed closely the observations for the most part but began to deviate afterwards. Despite the large discrepancies between the trends of observations and simulations, they differ only by 3%. Since our methane emissions rely on bottom-up emission inventories whose magnitudes can differ by more than 20% from the top-down estimates (Saunois et al., 2020), it is not difficult to introduce a 3% bias into our simulations. As methane is a long-lived**

**species, even a small deviation between the source and sink can be enough to cause our model to diverge from the actual values over time.**

| Modifications |
|---|
| **We added:**
"It is very probable that the extent of these downward trends in TOH has been exaggerated in our model because of the simulated $CH_4$ increasing too rapidly compared to in-situ observations. The overestimation of the upward trend in $CH_4$ in our model compared to in-situ observations could be caused by the biases (~3%) in sources minus sinks and/or the initial condition." |

L717-723: Does the global reduction of CO emissions contribute to the unexplained TOH trend?

| Response |
|---|
| **It can have an impact, but it is most likely overshadowed by methane too-rapidly rising in our model. We did not see a large negative trend in CO over central Africa (see Figure S15) to explain the positive trend in TOH.** |

---

## Author Comment (AC2)

Hydroxyl radical (OH) is the most important oxidant in the troposphere that governs the oxidation power, while its spatial-temporal patterns are yet to be clear at present. The authors of this study developed an integrated data- and model-driven approach to predict the convoluted response of TOH to its five proxies, which include NO2, HCHO, H2O, Trop O3, and Strat O3. They investigated the trends and drivers of global, hemispheric, and regional OH from 2005 to 2019. Overall, this study has provided interesting results, which will help deepen our understanding of the changes in global tropospheric OH over the past decades. I have several concerns about the method that could impact the robustness of the results.

1) Is there any method to evaluate the response of OH to different input parametrizations that were calculated by Eq. (5)? These semi-normalized sensitivities laid the foundation for understanding the impacts from different proxies/drivers on tropospheric OH. I would like to see some aspects of "evaluation" or at least a comparison with previous other studies.
* * *
**Response**

**We thank the reviewer for their constructive comments and the major point they raised about the evaluation of the response of OH to several parameters. We would like to address this comment in three ways: *the nature of this statistical approach vs. an explicit full chemistry model*, *qualitative agreement of the tendency of the sensitivity with theoretical expectations*, and *providing evidence of the capability of the framework at picking up non-linearities*.**

**The nature of this statistical approach vs. an explicit full chemistry model:**

**The limitation of statistical approaches in fully resolving "cause and effect" has been widely recognized in literature. Similarly, in the case of the ECCOH, we cannot entirely disentangle causation from correlation based solely on establishing a relationship between the distribution of OH and its influencers. As discussed in the paper and related references, the aim of the ECCOH framework is not to replicate all physical capabilities of a physics-based model, but rather to provide a first-order sensitivity experiment capable of qualitatively studying the effect of the biases in the input data on OH. This allows for more systematic strategies for full chemistry runs.**

**Robust evaluation of the sensitivities are mainly limited by the implicit nature of statistical methods. The implicit nature of statistical methods makes it difficult to identify a specific physiochemical process in a full chemistry model that is representative of the perturbation made to the OH parameterization. For example, perturbation of $NO_2$ in ECCOH represents the empirical relationship between OH and NO, which involves multiple processes, including $NO+HO_2$ ($RO_2$), $NO_2+OH$, the formation of ozone, aerosol HOx update, and radiation; thus, it is not restricted to specific chemical reaction. However, we do not know to what extent these processes are implicitly included in the perturbations. Similarly, the perturbation of HCHO does not necessarily represent the HCHO+OH reaction but rather where this compound can be used as a proxy for OH (discussed later).**

**Despite these challenges, it is important to recognize that the aim of ECCOH was not uniquely to improve the estimation of OH. Instead, the parameterization includes various input parameters (~ 27 inputs) so that the machine-learning algorithm could better understand the relationship between OH and its drivers for a wide range of atmospheric conditions. Because of this reason ECCOH has been able to represent OH distributions for extreme events (such as El Nino) that were not used during the training (Anderson et al., 2022, 2023, 2024). Without a proper**

establishment of the sensitivities (i.e., right OH prediction for a wrong reason), we would not have been able to reproduce OH distributions for such events.

ECCOH exhibits greater flexibility than some statistical approaches that use simpler assumptions (Valin et al., 2016; Murray et al., 2021; Wolfe et al., 2019; Pimlot et al., 2022), which may not fully capture complexities associated with the real atmosphere such as the effect of clouds and surface albedo or be applicable over high HOx conditions. For example, the global approximation of OH as function of reactive nitrogen formulated in Murray et al. (2021) may not be applicable for a wide range of atmospheric conditions. Anderson et al. (2022) carefully selected a vast number of OH-related parameters allowing us to better represent the response of OH to its drivers over both land and ocean and should be considered an improvement towards enhancing statistical-based OH studies.

**The sensitivities qualitatively agree with our theoretical expectations:**

$NO_2$ – It is believed to have positive feedback of reactive nitrogen on tropospheric OH based on both physics-based and statistical studies (Zhao et al., 2019, 2020, He et al. 2021, Chau et al., 2023; Naik et al., 2013; Murray et al., 2013; Strode et al., 2015; Nicely et al., 2018). Likewise, our perturbations in $NO_2$, as a surrogate for reactive nitrogen, causes TOH to increase. This increase happens in the tropospheric region. We will show in this response letter that non-linearities occur when we do the perturbations only at the surface layer where $NO_x$ is elevated because of different reactions such as ozone titration and $NO_2+OH$ which can reduce OH.

HCHO –We used this compound as a proxy (and not a driver) of OH following the studies of Valin et al. (2016) and Wolfe et al. (2019). The equation provided by Wolfe et al. (2019) follows:

$$[HCHO] = \frac{\alpha k'_{OH}[OH] + P_0}{j_{HCHO} + k_{HCHO+OH}[OH]}.$$

where the numerator is the production of HCHO from the oxidation of background VOCs, and the denominator is the sink of HCHO through both photolysis and the reaction with OH. In remote regions where $j_{HCHO} \gg k_{HCHO+OH}[OH]$, we can safely ignore the reaction of HCHO+OH, and assuming the minor source ($P_0$) to be zero, [HCHO] and [OH] become linearly correlated suggesting that if see a higher amount of HCHO, there has been more OH to oxidize background VOCs, assuming the slope stays constant (varies only by 5% based on Wolfe et al. (2016)). In high HOx regions, [HCHO] and [OH] becomes decoupled. This is what we essentially see from the perturbations in HCHO meaning the ECCOH response coincides with the theoretical expectation.

Stratospheric ozone – More stratospheric ozone hampers actinic flux leading to less production of $jO^1D$ resulting in a negative relationship between this quantity and OH.

Tropospheric ozone and water vapor – Both are primary source of OH and show positive feedback on OH. The magnitude of the positive sign can be influenced by the underlying surface albedo, clouds, or other implicit processes.

Now, the pivotal question is can we quantitatively assess these numbers? Since the perturbation of OH drivers in ECCOH are a snapshot of perturbing one variable without considering the response of unperturbed ones, we think it is challenging to replicate the identical experiments in a full chemistry model. Besides, the implicit nature of ECCOH makes it difficult to know exactly

which physiochemical processes we should collectively pick from a full chemistry model to compare with. Therefore, our confidence in perturbation has been mostly achieved through the "weight of evidence". As a result, all experiments done in our draft should be seen from a statistical and somewhat qualitative perspective.

**Providing evidence of the capability of the framework at picking up non-linearities**

This first reviewer raised a valid concern about the perturbation of $NO_2$. They pointed out that we did not have any negative values, but it is expected to see negative tendencies over extremely polluted regions where $NO_x$ can hamper OH levels. There are two reasons behind this. First, we focused on the tropospheric region where the majority of vertical grid boxes do not experience elevated $NO_x$ levels. Second, the M2GMI grid resolution is not spatially fine enough to fully resolve non-linear chemistry. However, this is not a major concern for the methane-CO-OH studies, as we intend to use ECCOH for climate studies at coarse resolution and not urban air quality applications. To demonstrate the capability of ECCOH at capturing negative sensitivities for more polluted regions, we applied the perturbation at the surface and calculated the changes in the surface OH mixing ratio. We indeed saw large negative values over polluted regions (shown later), which would be expected from a combined effect of ozone titration and $NO_2$+OH in a full chemistry model.

The capability of XGBoost at solving non-linear tendencies has been proven extensively (e.g., Johnson and Zhang, 2014: https://arxiv.org/pdf/1109.0887).

**Modifications**

To account for the reviewer's comment, we added more caveats throughout the paper and also included the perturbation experiments related to surface $NO_2$.

In the abstract, the description of the results is already qualitative. But to clarify, we added: "This innovative module helps efficiently predict the convoluted response of TOH to its drivers/proxies in a statistical way."

In section 2.2.3, right after introducing the method to get the perturbations:

"It is crucial to acknowledge that ECCOH has established an implicit relationship between OH and various input parameters statistically. These perturbations could involve a range of physiochemical processes that are challenging to fully decipher. For example, the perturbation of $NO_2$, acting as a surrogate of reactive nitrogen, involves chemical reactions that include reactive nitrogen like $NO$+$HO_2$ and $NO_2$+OH, ozone formation, aerosol $HO_x$ uptake, and radiation. Nonetheless, it may not be feasible to understand to what extent these processes have been represented by ECCOH. Therefore, the presented perturbations in this work should be viewed qualitatively."

In the results and discussion:

"Deciphering the precise chemical processes influencing the response of OH to $NO_2$ using a machine-learning approach is challenging. However, it is widely recognized that reactive nitrogen has positive feedback on tropospheric OH through increased $NO$+$HO_2$ and ozone (Murray et al., 2021; Zhao et al., 2020; He et al., 2021). Considering $NO_2$ as a surrogate for reactive nitrogen, similar tendencies are expected, as evident from the positive numbers from the sensitivity results obtained from offline calculations. The response of TOH to $NO_2$ displays a pronounced seasonal cycle stemming mainly from

photochemistry. It is believed to have some negative values for the sensitivity of OH to $NO_2$ for extremely polluted regions due to radical termination through $NO_2+OH$ or ozone titration (Nicely et al., 2018). While we have not identified any negative values in the tropospheric domain, we have observed significant negative values of OH when perturbing $NO_2$ at the model surface layer (Figure S*). This tendency highlights the ECCOH's ability to account for non-linearities."

[Figure]

**Figure S\*.** The sensitivity of surface OH to $NO_2$ perturbations in offline ECCOH in four different seasons.

To emphasize the relationship between HCHO and OH, we copy our discussion regarding **HCHO response map here:** "*The interplay between HCHO and OH is contingent on the intricate dynamics governing HCHO production from the oxidation of VOCs and methane and HCHO loss from various chemical pathways (Valin et al., 2016; Wolfe et al., 2019). In remote areas where $HO_x$ is low, the prevailing sink of HCHO is through photolysis. Conversely, in more polluted areas, the reaction of HCHO+OH emerges as a competing loss pathway. Assuming a steady-state approximation, which is a reasonable assumption for pristine areas, the photolysis loss of HCHO dominates over the reaction with OH, resulting in a linear relationship between HCHO and OH. In other words, high (low) HCHO concentrations are indicative of high (low) TOH. It is because of this that we use HCHO as a proxy of TOH in remote oceans regions. In regions characterized by heightened $HO_x$ levels, OH and HCHO become decoupled. Encouragingly, our implicit parametrization of OH has considerable skill at elucidating these intricate chemical tendencies; specifically, it reveals muted responses in regions with relatively tangible pollution levels, whereas positive responses are evident in oceanic regions. Like results obtained for $NO_2$, the response map has a seasonal cycle due to photochemistry.*" **The qualitative description of the response map for other factors had been provided in section 3.3.2.**

**In the conclusion section:**

"The development of an effective parameterization of OH, that is capable of integrating advanced satellite-based gas retrievals and improved weather forecast models enabled us to unravel the convoluted response of TOH to various parameters. Nonetheless, it is important to recognize some of the limitations associated with our work: first, the offline nature of the Bayesian data fusion algorithm makes the entire experiment blind to the interconnected responses of various compounds, such as ozone or aerosols, to adjustments to $NO_2$ and HCHO. Despite this limitation, our work has provided valuable

insights into the first-order effects of adjustments on TOH key inputs. This can help quickly identify areas where our prior knowledge is least reliable to simulate TOH. Second, the machine learning algorithm employed for parameterizing OH is implicit and its response to drivers/proxies is complex, making it difficult to quantitatively verify against full chemistry models. However, by including a vast number of parameters in the parameterization, Anderson et al. (2022) boosted its ability to understand the convoluted chemistry of OH. This has allowed for reproducing OH for events not included in the training dataset (Anderson et al., 2022, 2023, 2024)."

2) The input parameters were perturbed by the scaling factors of 1.1 and 0.9 in the ECCOH offline framework. What about the impacts of these assumed scaling factors? NOx and VOCs emissions might change by over 10% between 2005 and 2019, in both developed countries with advanced air pollution control and developing countries with rapid economic growth.

**Response**

**Thank you for your comment. All the results in the paper are derived from the online ECCOH framework, where various variables are dynamically changing over time (beyond 10%). The purpose of the offline perturbation was to approximate the response of OH to each driver/proxy in order to facilitate the interpretation of our results. The perturbation maps do not drive the results, but they can be used for sanity check. By using the first-order Taylor expansion, one can estimate the change of OH resulting from the changes in driver/proxy.**

$$\Delta OH = S_{driver}*(\Delta X/X),$$

**where $S_{driver}$ is the first-order perturbation and X is a driver concentration. This means that we could approximate the third column in Figure 5 by multiplying the sensitivity map by the relative changes of $NO_2$. The sensitivity is independent of how the driver changes as it tries to approximate the first-order derivative. As mentioned, the ultimate results are driven by multiple variables changing in the $CH_4$-CO-OH system. Even though in the model framework, higher-order changes across all variables are affecting the results, having a first-order approximation is enough to understand how ECCOH works. When we applied the above equation to values in Figure 5, we noticed that the differences in $\Delta OHs$ are very close to what we get from the online model, indicating that it is a reasonable approximation for sanity check.**

[Figure]

**Modifications**

**To clarify that we do not rely on these perturbations to generate the modeling results, we added:**

"To elucidate the response of OH to different input parametrizations, such as $NO_2$, HCHO, and $O_3$, we determine the semi-normalized sensitivities through a traditional finite difference method:

$$SOH_i = \frac{[OH]_i^{110\%} - [OH]_i^{90\%}}{0.2} \tag{5}$$

where $[OH]_i^{110\%}$ and $[OH]_i^{90\%}$ are OH concentrations from perturbing input parameters (*i*) by 1.1 and 0.9 scaling factors in the ECCOH offline framework (Anderson et al., 2022). These calculations are solely used to better understand why OH changes in a particular way relative to the changes in its drivers. In our online modeling framework, OH is simultaneously affected by the dynamic changes of various variables considered in the parametrization of OH."

3) I feel that the trends in OH present in this study are substantially impacted by the trends of NO2 and HCHO, which were constrained by OMI. What about the impacts of uncertainties in the trends of OMI observations on the OH estimates? Especially for HCHO, I have seen some papers showing that there were systematic errors in the global and latitudinal trends.

**Response**

The main purpose of using Bayesian data fusion is to make sure that only reliable satellite observations are used to make adjustments to the OH input parameterization. Additionally, the algorithm takes into account the differences between the MERRA2GMI fields and the data from the satellites. The initial concept is defined by the Kalman gain:

$\mathbf{BH}^T(\gamma \mathbf{HBH}^T + \mathbf{E})^{-1}$

And the second part by:

$(\mathbf{Y} - \mathbf{HX}_b)$

Their definitions can be found in Section 2.2.1. The multiplication of these two terms determines the adjustments applied to the ECCOH inputs. If the satellite observations are uncertain (low Kalman gain) and/or the difference between the satellite observations and the a priori (here M2GMI) is small, the results won't depend on the satellite information. In our draft, we observe both scenarios: OMI NO$_2$ provides reasonable information over cities, however the M2GMI NO$_2$ trends are not substantially different from the OMI (except for China). So, the linear trend results are largely influenced by the M2GMI information. We exclusively investigated this pattern in section *3.3.3, OMI contributions to TOH trends*. For instance, we cannot gain a high amount of information from OMI HCHO over oceans, so regardless of the difference between M2GMI and OMI, the Bayesian algorithm does not suggest large adjustments.

However, we did not analyze the accuracy of the linear trends of OMI NO$_2$ and HCHO on a global scale. This is because we do not have a large number of long-term records of sky-radiance spectrometers that cover the period from 2005 to 2019. This limitation will not be an issue for future research, as many surface networks are expanding (e.g., FTIR, MAX-DOAS, Pandora). Nevertheless, we qualitatively compared the sign of our trends with several reputable studies.

To our best knowledge, our paper is one of the first attempts to leverage the newest version of OMI HCHO (v4) that has shown tremendous improvements with respect to in-situ measurements (Ayazpour et al., (2023): https://agu.confex.com/agu/fm23/meetingapp.cgi/Paper/1407690). In particular, the radiance information has become more stable over time in V4, and the latitudinal biases have been removed based on Nowlan et al. (2023). The first author has closely worked with that team prior to joining GSFC and is fully aware of the new improvements.

4) Lines 498-500, the authors said "The correction factors, however, worsen the trends over the southeast US and Canada. This is essentially due to the use of the fractional errors in the a priori making the OMI corrections more impactful (i.e., higher Kalman gain) in summertime than in wintertime." I did not understand what these sentences meant. Why did more impactful OMI corrections worsen the trends over the southeast US and Canada?

| Response |
| --- |
| **Due to the incorporation of data point errors in our trend analysis calculation, the data fusion algorithm works to reduce the errors of data points during periods of strong OMI HCHO signals (warmer seasons). This means that instead of assuming a constant 50% error, we now have varying errors with lower values during summertime and higher values (close to 50%) during wintertime (due to low SNR from OMI). As a result, we believe that this has caused some disagreements between OMI and the posterior estimates, particularly in statistically insignificant areas such as the southeast US, which experience large interannual and interdecadal variabilities. These variable errors can lead to different outcomes from the linear trend equation. When we don't take errors into account, the posterior results are similar to the prior ones, but we believe it's crucial to consider the errors.** |

| Modifications |
| --- |
| **To better clarify and tone down this tendency we rewrote:**

"The posterior estimates better line up with the OMI trends, especially over the Amazon, India, and Central Africa (Text S3). The correction factors, however, worsen the trends over the southeast US and Canada. One possible explanation for this may be the varying errors from the data fusion algorithm, which tend to be reduced more in summertime than in wintertime due to the larger OMI HCHO signal. This results in some degree of inconsistencies of the linear trend over these regions with larger interannual and interdecadal variabilities.". |